# Improved Invariant Learning for Node-level Out-of-distribution Generalization on Graphs

## Abstract

Enhancing OOD generalization on graph data is a recent hot research topic. Among this, node-level OOD generalization remains an underexplored and challenging subject. The difficulty of node-level OOD tasks lies in the fact that representations between nodes are coupled through edges, making it difficult to characterize distribution shifts and capture invariant features. Furthermore, in practice, environment labels for nodes are typically expensive to obtain, rendering invariant learning strategies based on environment partitioning infeasible. By establishing a theoretical model, we highlight that even with ground-truth environment partitioning, classical invariant learning methods like IRM and VREx designed for independently distributed training data will still capture spurious features when the depth of the GNN exceeds the width of a node's causal pattern (i.e., the invariant and predictive neighboring subgraph). Intriguingly, however, we theoretically and empirically find that by enforcing the Cross-environment Intra-class Alignment (CIA) of node representations, we can remove the reliance on these spurious features. To harness the advantages of CIA and adapt it on graphs, we further propose Localized Reweighting CIA (**LoRe-CIA**), which does not require environment labels or intricate environment partitioning processes. Leveraging the neighboring structural information of graphs, LoRe-CIA adaptively selects node pairs that exhibit large differences in spurious features but minimal differences in causal features for alignment, enabling better elimination of spurious features. The experiments on the GOOD benchmark show that LoRe-CIA achieves optimal OOD generalization performance on average.

## 1 Introduction

Generalizing to unseen testing distributions that differ from the training distributions, also known as achieving Out-Of-Distribution (OOD) generalization, is one of the key challenges in machine learning. In OOD scenarios, the performance of models often deteriorates due to the distribution discrepancy between training and testing data. Among various OOD tasks, enhancing OOD generalization on graph data is an emerging research direction that is garnering increasing attention (Chen et al., 2022; Zhu et al., 2021; Wu et al., 2022; Li et al., 2023; Wu et al., 2021; Liu et al., 2023; Li et al., 2022a;b; Yang et al., 2022; Chen et al., 2023; Buffelli et al., 2022; Tang & Liu, 2023). These works can be broadly categorized into graph-level and node-level OOD generalization. This paper focuses on the latter task, which has yet to be fully explored.

For addressing OOD generalization, invariant learning is one of the crucial strategies. The idea of invariant learning is to capture causal features that remain consistent across different environments and have predictive power, thereby maintaining performance in the presence of distributional shifts. Numerous invariant learning methods have been proposed to tackle OOD problems in CV and NLP tasks (Arjovsky et al., 2020; Krueger et al., 2021; Bui et al., 2021; Rame et al., 2022; Shi et al., 2021; Mahajan et al., 2021; Wang et al., 2022; Yi et al., 2022). Nevertheless, transferring these methods to node classification tasks on graphs is not straightforward. This is because graph data is distinct from Euclidean data like images and text. Beyond node features, it incorporates topological structure information, making the task of characterizing the specific nature of distribution shifts and capturing invariant features more challenging. Compared to graph classification, the node classification task

is more complex due to the existence of edges between samples, causing the samples not to be independently distributed. Additionally, environment labels in node-level tasks are difficult to obtain (Wu et al., 2021), further preventing the application of previous invariant learning methods based on environment partitioning.

To investigate the performance of invariant learning objectives when using message-passing Graph Neural Networks (GNNs), along with the presence of shifts in both the graph's topological structure and node features, we establish a theoretical model (Section 3.1). We discover that IRM (Arjovsky et al., 2020) and VREx (Krueger et al., 2021), two classic OOD methods originally designed for non-graph data (where samples are independently distributed), tend to learn spurious features in node-level OOD tasks. The latter, which aims to minimize the variance across training environments, is the loss function adopted by many graph OOD methods (Wu et al., 2021; Yu et al., 2023; Wu et al., 2022; Li et al., 2023; 2022b). This highlights the need to seek better objectives for invariant learning on graphs. Surprisingly, we theoretically find that enforcing Cross-environment Intra-class Alignment (CIA) of node representations, which was proposed for domain generalization (Mahajan et al., 2021), can remove spurious features in node-level OOD scenarios. However, we also identified the issue of causal feature collapse resulting from excessive alignment. To mitigate the above problems and harness the benefits of the CIA, we propose LoRe-CIA, designed to adapt CIA to graph tasks and achieve improved node-level OOD generalization.

Our contribution is as follows:

1. We construct a theoretical toy model to analyze the performance of different invariant learning methods on node-level OOD tasks and reveal that IRM and VREx, which have been proven to be effective when training on independently distributed samples, rely on spurious features on the graph data, where training samples (nodes) are correlated (Section 3.2).

2. Based on our theoretical model, we find that Cross-environment Intra-class representation Alignment (CIA) can learn invariant representations even on graph data (Section 3.3). We empirically validate the effectiveness of CIA (Section 5.2). However, we identify the representation collapse caused by too strong CIA regularization, which will lead to the degradation of OOD accuracy (Section 4.1).

3. We propose a new method that requires no environment labels and avoids complex environmental partitioning processes, utilizing localized patterns to find node pairs with large spurious feature differences and small causal feature differences for alignment, achieving better spurious feature elimination and better OOD generalization (Section 4.2).

4. We evaluate the proposed method on a popular graph OOD benchmark GOOD (Gui et al., 2022) and demonstrate that it achieves state-of-the-art performance and even outperforms many methods that use ground-truth environment labels (Section 5.2).

## 2 PROBLEM FORMULATION

In the node-level graph OOD task, we are given a single training graph $\mathcal{G} = (A, X, Y)$ containing $N$ nodes $\mathcal{V} = \{v_i\}_{i=1}^N$ from multiple training environments $e \in \mathcal{E}_{\text{tr}}$. $A \in \{0, 1\}^{N \times N}$ is the adjacency matrix, $A_{i,j} = 1$ iff there is an edge between $v_i$ and $v_j$. $X \in \mathbb{R}^{N \times D}$ are node features. The $i$-th row $X_i \in \mathbb{R}^D$ represents the feature of $v_i$. $Y \in \{0, 1, ..., C - 1\}^N$ are the labels, $C$ is the number of the classes. $E \in \mathbb{R}^N$ are environment labels that are usually unattainable. Denote the subgraph containing nodes of environment $e$ as $\mathcal{G}^e = (A^e, X^e, Y^e)$. $\mathcal{G}^e$ follows the distribution $p_e$. Suppose the unseen test environments are $e' \in \mathcal{E}_{\text{te}}$. Denote the test graphs as $\mathcal{G}^{e'} = (A^{e'}, X^{e'}, Y^{e'})$ which follow the test distribution $p_{e'}$. The test distributions are different from the training distributions, i.e., $p_{e'} \neq p_e, \forall e' \in \mathcal{E}_{\text{te}}, \forall e \in \mathcal{E}_{\text{tr}}$.

Denote the GNN model parameterized by $\Theta$ as $f_\Theta : (\mathcal{A}, \mathcal{X}) \to \mathcal{Y}$ that maps a graph from input space to label space. The goal of OOD generalization is to minimize the risk over test distributions:

$$\min_\Theta \max_{e'} \mathbb{E}_{e'} \left[ \mathcal{L} \left( f_\Theta(\mathcal{G}(A^e, X^e)), Y^e \right) \right], \tag{1}$$

where $\mathcal{L}$ is a loss function.

# 3 A THEORETICAL MOTIVATION

In this section, we will construct a toy theoretical model to analyze the performance of several OOD methods in node-level OOD classification problems.

## 3.1 THEORETICAL MODEL SETUP

Let's consider a simple case that each node $v$ in environment $e$ has a 2-dim feature $[x_v^1, x_v^{2e}]^\top$, $N^e$ is the number of samples in $e$. The node features of all samples in $e$ are denoted as $X_1 \in \mathbb{R}^{N^e \times 1}$ and $X_2^e \in \mathbb{R}^{N^e \times 1}$ corresponding to $x_v^1$ and $x_v^{2e}$. $Y^e \in \mathbb{R}^{N^e \times 1}$ are the labels[1]. Let $A^e \in \{0,1\}^{N^e \times N^e}$ and $D^e$ be the adjacency matrix and the diagonal degree matrix respectively, where $D_{ii}^e = \sum_{j=1} A_{ij}^e$. Denote the normalized adjacency matrix as $\tilde{A}^e = (D^e + I_{N^e})^{-\frac{1}{2}}(A^e + I_{N^e})(D^e + I_{N^e})^{-\frac{1}{2}}$, $I_{N^e}$ is the identity matrix. The data generation process for environment $e$ is

$$Y^e = \tilde{A}^{e^k} X_1 + n_1, \quad X_2^e = \tilde{A}^{e^m} Y^e + n_2 + \epsilon^e = \tilde{A}^{e^s} X_1 + \tilde{A}^{e^m} n_1 + n_2 + \epsilon^e, \quad (2)$$

where $n_1 \in \mathbb{R}^{N^e \times 1}$ and $n_2 \in \mathbb{R}^{N^e \times 1}$ are vectors representing noise with each dimension independently following the standard normal distribution. $\epsilon^e \in \mathbb{R}^{N^e \times 1}$ is an environment spurious variable. $\epsilon_i^e$ (each dimension of $\epsilon^e$) is a random variable that are independent for $i = 1, ..., N^e$. We assume the intra-environment expectation of the environment spurious variable is $\mathbb{E}_{\epsilon_i \sim p_e}[\epsilon_i] = \mu^e \in \mathbb{R}$ since spurious features are consistent in a certain environment. We further assume the cross-environment expectation $\mathbb{E}_e[\epsilon^e] = \mathbf{0}$ and cross-environment variance $\mathbb{E}_e[\epsilon_i^e] = \sigma^2$, $i = 1, ..., N^e$ for simplicity. Note that $X_1$ and $X_2^e$ denote invariant features (causing $Y^e$) and spurious features (effects of $Y$) that vary with environments. $k$ and $l$ are the depth of the generation process of invariant and spurious features respectively. We also have the following assumption about the stability of the causal feature across environment in our motivation example:

**Assumption 3.1.** *(Local stability of the causal patterns) The causal feature is stable across environments for every class c:* $\forall e$, $(\tilde{A}^{e^k} X_1)_{[c][i_c^e]} \in \mathbb{R}^{1 \times 1}$, $i_c^e = 1, ..., N_c^e$ *are the same, where* $(\tilde{A}^{e^k} X_1)_{[c][i_c^e]}$ *are the elements of* $(\tilde{A}^{e^k} X_1)_{[c]} \in \mathbb{R}^{N_c^e \times 1}$, *which corresponds to the causal features of class c in environment e,* $N_c^e$ *is the number of such samples.*

**Remark.**

1. In this toy model, the distribution shift is caused by both the changes of topological structures ($A^e$) and node features ($X_2^e$). This is the general case of real-world OOD graphs. Note that although the global structures vary across environments, we assume the causal feature of each class is locally stable (Assumption 3.1).

2. We extend the theoretical model to a more general case than the toy model in EERM (Wu et al., 2021): we consider multi-layer data generation processes and GNNs. Nevertheless, the data-generating process considered in EERM is a simple ego-graph of a centered node with its 1-hop neighbors, and they use a 1-layer GNN.

Now we introduce the toy GNN used in the following analysis. Consider a $L$-layer GNN $f$ parameterized by $\Theta = \left\{ \theta_1, \theta_2, \theta_1^{1(l)}, \theta_1^{2(l)}, \theta_2^{1(l)}, \theta_2^{2(l)} \right\}$, $l = 1, 2, ..., L-1$:

$$\begin{aligned}
f_\Theta(A, X) &= H_1^{(L)}\theta_1 + H_2^{(L)}\theta_2, \\
H_1^{(l)} &= \theta_1^{1(l-1)} \bar{A} H_1^{(l-1)} + \theta_1^{2(l-1)} \bar{I}_{N^e} H_1^{(l-1)}, \quad l = 2, 3, ..L \\
H_2^{(l)} &= \theta_2^{1(l-1)} \bar{A} H_2^{(l-1)} + \theta_2^{2(l-1)} \bar{I}_{N^e} H_2^{(l-1)}, \quad l = 2, 3, ..L, \\
H_1^{(1)} &= X_1, \\
H_2^{(1)} &= X_2
\end{aligned} \quad (3)$$

where $\bar{A} = (D + I_{N^e})^{-\frac{1}{2}} A (D + I_{N^e})^{-\frac{1}{2}}$ and $\bar{I}_{N^e} = (D + I_{N^e})^{-\frac{1}{2}} I_{N^e} (D + I_{N^e})^{-\frac{1}{2}}$. In this simple $L$-layer GNN, we omit the activation function and simplify the weight matrix to four scalar

---

[1]Different from the definition in Section 2, we denote a vector $V \in \mathbb{R}^D$ in the matrix form of $V \in \mathbb{R}^{D \times 1}$.

parameters $\theta_1^{1\,(l)}$, $\theta_2^{1\,(l)}$, $\theta_1^{2\,(l)}$, $\theta_2^{2\,(l)}$ in each layer. Suppose $L \geq k$. $\theta_1^{1\,(l)}$, $\theta_2^{1\,(l)}$ are weights for aggregating features from neighboring nodes and $\theta_1^{2\,(l)}$, $\theta_2^{2\,(l)}$ are weights for features of a centered node. This toy GNN can be seen as a simplified GAT (Graph Attention Network, (Veličković et al., 2018)). In this GNN, we simplify the classifier to an identity mapping so the featurizer $\phi$ is $f$.

**Remark.** We assume $L \geq k$ to ensure the model has the capacity to learn causal features of nodes. To verify this assumption, we train GCNs with different numbers of layers to predict the ground-truth labels. We find that on real-world large scale datasets, the depth of the generation is no more than 4 in most cases (reults are in Appendix C.3). Note that this scenario was not considered in the theoretical motivation model of EERM (Wu et al., 2021) (where they considered a 1-layer model). Later we will show that this basic setup will lead to unexpected failure of IRMv1 and VREx.

Consider a regression problem that we aim to minimize the MSE loss over all environments $\mathbb{E}_e[R(e)] = \mathbb{E}_e\left[\mathbb{E}_{n_1,n_2}\left[\|f_\Theta(A^e, X^e) - Y^e\|_2^2\right]\right]$. The optimal parameter set $\Theta^*$ is

$$
\begin{cases}
\theta_1 = 1 \\
\theta_2 = 0 \quad \text{or} \quad \exists l \in \{1, ..., L-1\} \text{ s.t. } \theta_2^{1\,(l)} = \theta_2^{2\,(l)} = 0 \\
\theta_1^{1\,(l)} = 1, \theta_1^{2\,(l)} = 1, \quad l = L-1, ..., L-k+1 \\
\theta_1^{1\,(l)} = 0, \theta_1^{2\,(l)} = 1, \quad l = L-k, L-k-1, ..., 1
\end{cases}
\tag{4}
$$

which only uses the invariant features $X_1$ for prediction.

## 3.2 FAILURE CASES OF SOME PREVIOUS OOD ALGORITHMS

In this section, we present two failure cases on the node-level OOD task: optimizing VREx (Krueger et al., 2021) and IRMv1 (Arjovsky et al., 2020) induces a model that relies on spurious features $X_2^e$ to predict, which will lead to poor OOD generalization performance. Subsequently, we will analyze the reasons for the failure and attempt to find solutions.

### 3.2.1 FAILURE CASE OF VREX

First, we will show minimizing the risk variance (VREx) is an ill-posed problem and may lead to a solution that relies on spurious features. The proof is in Appendix E.1.1.

**Proposition 3.2.** *(VREx will use spurious features)* *The objective* $\min_\Theta \mathcal{L}_{VREx} = \mathbb{V}_e[R(e)]$ *has non-unique solutions, and when part of the model parameters* $\{\theta_1^{1\,(l)}, \theta_1^{2\,(l)}, \theta_2^{1\,(l)}, \theta_2^{2\,(l)}\}$ *take the values*

$$
\Theta_0 = \begin{cases}
\theta_1^{1\,(l)} = 1, \theta_1^{2\,(l)} = 1, \quad l = L-1, ..., L-s+1 \\
\theta_1^{1\,(l)} = 0, \theta_1^{2\,(l)} = 1, \quad l = L-s, L-s-1, ..., 1 \\
\theta_2^{1\,(l)} = 0, \theta_2^{2\,(l)} = 1, \quad l = L-1, ..., 1
\end{cases},
\tag{5}
$$

*for some* $0 < s < L$, $\theta_1$ *and* $\theta_2$ *have four sets of solutions of the cubic equation:*

$$
\begin{cases}
(3c_1\theta_1\theta_2 + c_1(\theta_2)^2 - 2c_6\theta_2)\sigma^2 - \mathbb{E}_e[N^e(2c_1(\theta_1 + \theta_2) - c_6)]\sigma^2\theta_2 + c_7 = 0 \\
(\mathbb{E}_e[N^e(2c_1(\theta_1 + \theta_2) - c_6)]\sigma^2\theta_2 - c_7)(c_3\theta_2 - c_4) - [c2(\theta_1 + \theta_2) - c_5](\theta_2)^2 = 0
\end{cases}.
\tag{6}
$$

*where* $c_1, c_2, ..., c_7$ *are some constants.*

According to Proposition 3.2, since $\theta_2 = 0$ is not a solution to the above equation set, VREx will inevitably use spurious features $X_2^e$ [2].

### 3.2.2 FAILURE CASE OF IRMV1

Next, we give another proposition showing that optimizing IRMv1 could also fail on the graph OOD task. The proof is in Appendix E.1.2.

---

[2]Note that Wu et al. (2021) proves $\min_\Theta \mathbb{V}_e[R(e)]$ will min $I(y, e|z)$, where $q(z|x)$ is the induced distribution by encoder $\phi$. This seems to conflict with Proposition 3.2. This may be because the upper bound $I(y, e|z) \leq D_{\mathrm{KL}}(q(y|z)\|\mathbb{E}_e[q(y|z)]) \leq \mathbb{V}_e[R(e)]$ is not tight.

**Proposition 3.3.** *(IRMv1 will use spurious features) The objective* $\min_\Theta \mathcal{L}_{IRMv1} = \mathbb{E}_e[\|\nabla_{w|w=1.0} R(e)\|^2]$ *has a solution that uses spurious features:*

$$
\begin{cases}
\theta_1 = \dfrac{\mathbb{E}_e\left\{(\tilde{A}e^s X_1)^\top(\tilde{A}e^k X_1)\left[\mathbf{1}^\top\tilde{A}e^s X_1 + (\tilde{A}e^k X_1)^\top(\tilde{A}e^k\mathbf{1})\right] + (1+\sigma^2)(\tilde{A}e^s X_1)^\top\mathbf{1}(\tilde{A}e^k X_1)^\top\mathbf{1}\right\}}{(2+\sigma^2)(\mathbb{E}_e[\tilde{A}e^s]X_1)^\top\mathbf{1}} \\[4mm]
\theta_2 = \dfrac{\mathbb{E}_e\left\{(\tilde{A}e^s X_1)^\top(\tilde{A}e^s X_1)[\mathbf{1}^\top(\tilde{A}e^k\mathbf{1})] + (\tilde{A}e^s X_1)^\top(\tilde{A}e^k X_1)(\mathbf{1}^\top\tilde{A}e^s X_1)\right\}}{(2+\sigma^2)(\mathbb{E}_e[\tilde{A}e^s]X_1)^\top\mathbf{1}}
\end{cases}
\tag{7}
$$

*when* $\{\theta_1^{1}{}^{(l)}, \theta_1^{2}{}^{(l)}, \theta_2^{1}{}^{(l)}, \theta_2^{2}{}^{(l)}\}$ *take the special values* $\Theta_0$ *(defined in Equation (5)).*

Let's intuitively explain the failure cases in detail. When $L \geq k$, and when the lower-layer parameters of the GNN $\theta_1^{1}{}^{(l)}, \theta_2^{1}{}^{(l)}, \theta_1^{2}{}^{(l)}, \theta_2^{2}{}^{(l)}$ take the specific values $\Theta_0$, we have

$$
H_1^{(L)} = \frac{\partial H_1^{(L)}}{\partial \theta_1^{i}{}^{(l)}} = \tilde{A}e^s X_1, \quad i = 1, 2, \quad l = 1, ..., L-1,
\tag{8}
$$

and

$$
H_1^2(L) = \frac{\partial H_2^{(L)}}{\partial \theta_2^{i}{}^{(l)}} = \tilde{A}e^{k+m} X_1, \quad i = 1, 2, \quad l = 1, ..., L-1,
\tag{9}
$$

holds for every environment $e$. Thus, we get

$$
\begin{aligned}
\frac{\partial \mathcal{L}}{\partial \theta_1} &= \frac{\partial \mathcal{L}}{\partial (H_1^{(L)}\theta_1)} \frac{\partial (H_1^{(L)}\theta_1)}{\partial \theta_1} = \frac{\partial \mathcal{L}}{\partial (H_1^{(L)}\theta_1)} H_1^{(L)} \\
&\stackrel{(*)}{=} \frac{\partial \mathcal{L}}{\partial (H_1^{(L)}\theta_1)} \frac{\partial (H_1^{(L)})}{\partial \theta_1^{i}{}^{(l)}} = \frac{\partial \mathcal{L}}{\partial \theta_1^{i}{}^{(l)}} \frac{1}{\theta_1}, \quad i = 1, 2, \quad l = 1, ..., L-1
\end{aligned}
\tag{10}
$$

$(*)$ is because of Equation (9). Therefore, $\frac{\partial \mathcal{L}}{\partial \theta_1} = 0 \Rightarrow \frac{\partial \mathcal{L}}{\partial \theta_1^{i}{}^{(l)}} = 0$. The same is true for $\frac{\partial \mathcal{L}}{\partial \theta_2}$ and $\frac{\partial \mathcal{L}}{\partial \theta_2^{i}{}^{(l)}}$. This means the solution of the top-level parameters $\theta_1$ and $\theta_2$ of the GNN will only be constrained by two equations, $\frac{\partial \mathcal{L}}{\partial \theta_1} = 0$ and $\frac{\partial \mathcal{L}}{\partial \theta_2} = 0$, rather than be constrained by all gradient functions $\frac{\partial \mathcal{L}}{\partial \theta_i^j} = 0$, $i = 1, 2$. By analyzing the specific loss of VREx and IRMv1, we conclude that they will induce a non-zero $\theta_2$.

From the above analysis, we know that the key to avoid relying on spurious features is appropriately regularizing the gradients for top-level parameters, so that the spurious components are discarded. Therefore, we consider explicitly introducing the elimination of spurious features into the loss. In the next section, we will point out our finding: aligning cross-environment intra-class features is an effective strategy for graph node-level OOD tasks.

### 3.3 Cross-environment Intra-class Alignment Can Get Rid of Spurious Features

With a slight abuse of the notation, denote $\phi_\Theta(A^e, X^e)_{[c][i]}$ as the representation of a node $v_i$ which has class $c$ and environment label $e$. Now we will give a proposition demonstrating that aligning the representations of nodes of the same class from different environments $e$ and $e'$ can learn the parameters without relying on spurious features. This is similar to a popular objective for domain generalization (Mahajan et al., 2021) where they used a contrastive loss to match the representations of samples with the same causal features. Since the objective we consider here does not push representations of different classes apart while Mahajan et al. (2021) did, we rename this method CIA (Cross-environment Intra-class Alignment).

The CIA's objective is:

$$
\begin{aligned}
\min_\Theta \quad & \mathbb{E}_e\left[\mathcal{L}(f(A^e, X^e))\right] \\
\text{s.t.} \quad \min_\Theta \quad & \mathcal{L}_{\text{CIA}} = \mathbb{E}_{\substack{e,e' \\ e \neq e'}} \mathbb{E}_c \mathbb{E}_{\substack{i,j \\ (i,j)\in\Omega_c^{e,e'}}} \left[\mathcal{D}(\phi_\Theta(A^e, X^e)_{[c][i]}, \phi_\Theta(A^e, X^{e'})_{[c][j]})\right]
\end{aligned}
\tag{11}
$$

where $\Omega_c^{e,e'} = \{(i,j) | i \neq j \wedge Y_i = Y_j = c \wedge E_i = e, \ E_j = e'\}$.

**Proposition 3.4.** *Optimizing the CIA objective will lead to the optimal solution* $\Theta^*$:

$$
\begin{cases}
\theta_1 = 1 \\
\theta_2 = 0 \quad or \quad \exists l \in \{1, ..., L-1\} \ s.t. \ \theta_2^{1(l)} = \theta_2^{2(l)} = 0 \\
\theta_1^{1(l)} = 1, \theta_1^{2(l)} = 1, \quad l = L-1, ..., L-k+1 \\
\theta_1^{1(l)} = 0, \theta_1^{2(l)} = 1, \quad l = L-k, L-k-1, ..., 1
\end{cases}
. \tag{12}
$$

The proof is in Appendix E.1.3. The failure cases in Proposition 3.2 and 3.3 do not work for CIA since solving $\frac{\partial \mathcal{L}_{\text{CIA}}}{\partial \theta_1} = 0$ and $\frac{\partial \mathcal{L}_{\text{CIA}}}{\partial \theta_2} = 0$ alone leads to the conclusion that the spurious parameters must be 0. When CIA objective is well optimized, further minimize $\mathbb{E}_e\left[\mathcal{L}(f(A^e, X^e))\right]$ will induce the parameters informative of the labels.

# 4 LOCALIZED REWEIGHTING CIA: AN ENVIRONMENT-LABEL FREE ADAPTATION TO NODE-LEVEL OOD TASKS

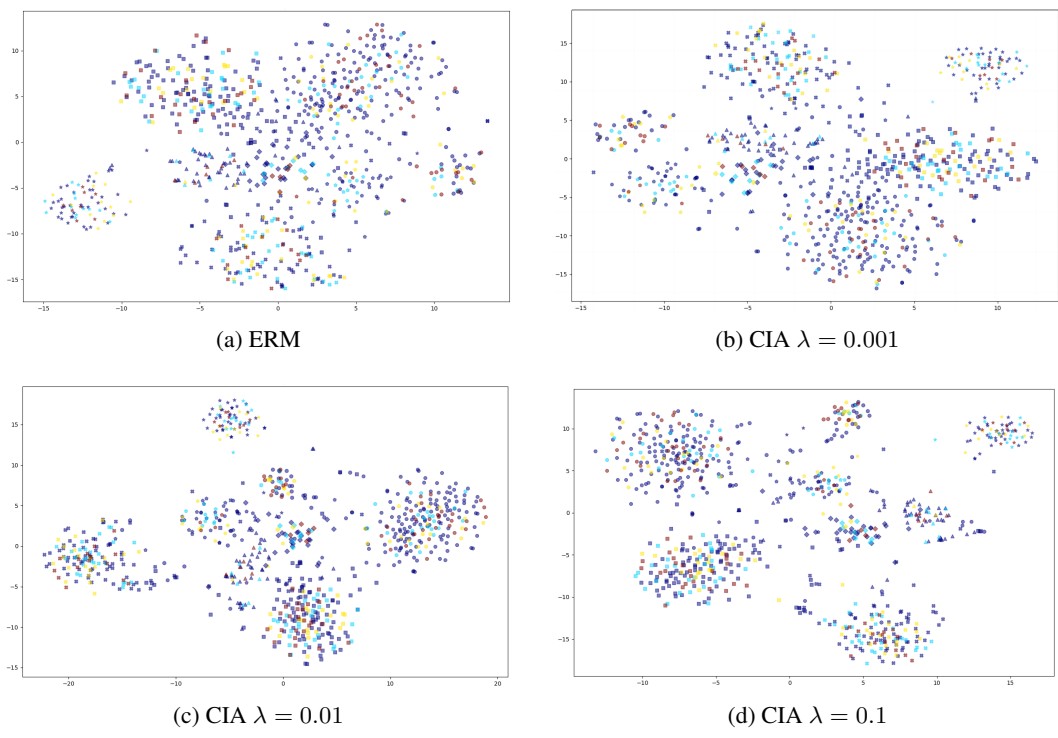

(a) ERM                      (b) CIA $\lambda = 0.001$

(c) CIA $\lambda = 0.01$        (d) CIA $\lambda = 0.1$

Figure 1: Visualization of the representations learned by ERM and CIA on Cora (concept). Classes and environments are distinguished by shape and color, respectively. Larger $\lambda$ means stronger CIA regularization. CIA can better separate different classes. The representation distribution of each class collapses into a compact region when CIA regularization is strong.

## 4.1 FEATURE COLLAPSE DUE TO EXCESSIVE ALIGNMENT

We have shown that CIA is a more effective proposal for OOD node classification than VREx and IRMv1, however, it still has several limitations in practical applications. The first problem is CIA requires knowledge of the ground-truth environmental labels (like many previous classic OOD algorithms), which is challenging to obtain in most node classification tasks (Wu et al., 2021; Liu et al., 2023). Another issue is the collapse of causal features. In the motivation model, we assume the causal feature for a class takes one specific value. However, in practical situations, causal features also exhibit diversity. If the causal features of the samples aligned by CIA vary greatly, the model might optimize the CIA loss by reducing the distance between the causal representations of the two

samples, rather than solely eliminating the discrepancies between the spurious features, leading to causal representation collapse. As shown in Figure 1, when we add the CIA loss as a regularizer to the ERM objective with weight $\lambda$, the learned representations become increasingly compact for each class as $\lambda$ increases. However, as indicated in Table 1, the test performance declines when $\lambda$ is large. This suggests that CIA may lose the diverse information that is useful for predictions.

Table 1: OOD test accuracy (%) of ERM and CIA on CBAS. Larger $\lambda$ means stronger alignment. **Message 1:** For regular CIA, too strong alignment regularization will lead to the collapse of the causal feature, resulting in sub-optimal performance. **Message 2:** LoRe-CIA will alleviate the feature collapse problem. We choose the neighboring hops to be 3 for LoRe-CIA. The values in parentheses are standard deviations, best results of each line are shown in **bold**.

| Dataset | Algorithm | $\lambda = 0.0005$ | $\lambda = 0.005$ | $\lambda = 0.05$ | $\lambda = 0.1$ | ERM |
|---|---|---|---|---|---|---|
| CBAS (covariate) | CIA | **78.57(1.43)** | 77.62(0.83) | 73.34(0.67) | 70.00 (1.43) | 78.57(2.02) |
| | LoRe-CIA | 79.52(0.67) | **81.43(0.00)** | 79.52(1.35) | 80.00(1.17) | |
| CBAS (concept) | CIA | 84.29(0.72) | **85.00(2.14)** | 83.81(2.30) | 80.48 (1.48) | 82.14(1.17) |
| | LoRe-CIA | 81.90(0.89) | 82.14(1.75) | 85.47(0.33) | **85.72(1.54)** | |

### 4.2 Proposed Localized Reweighting CIA

To address the above issues, we aim to identify node pairs for CIA with significant differences in spurious features and small differences in causal features. In this way, during alignment, the model would be more inclined to eliminate spurious features rather than causal ones. To this end, we propose an environment-label free reweighting CIA strategy to utilize node neighborhood features to screen for such pairs. Now we describe our method.

The first assumption we rely on is that the rate of change of a node's spurious features w.r.t. spatial location on the graph is faster than that of the invariant features within a certain range of hops. We verify this intuition on real-world datasets Arxiv and Cora in Appendix C.4. Therefore, we choose to align nodes of the same class within a certain number of hops, as two same-class nodes that are too far apart on the graph might also have substantial differences in their causal features. The second intuition is that the label distribution of a node $v$'s different-class neighbors can reflect the distribution of spurious features of $v$. This is empirically verified and discussed in Appendix C.5. We resort to this observation to find node pairs with large differences in spurious features. We estimate the difference in spurious distribution by computing the difference in the number of heterophilous neighbors. Similarly, the difference in the causal feature can be estimated as the number of neighboring nodes with the same label as $v$. Now we are ready to present the formal objective of the proposed method: **LoRe-CIA** (Localized Reweighting CIA):

$$
\begin{aligned}
\min_{\Theta} \quad & \mathbb{E}_e[\mathcal{L}(f(A^e, X^e))] \\
\text{s.t.} \quad \min_{\Theta} \quad & \mathbb{E}_c \mathbb{E}_{\substack{i,j \\ (i,j) \in \Omega_c(t)}} \left[ w_{i,j} \mathcal{D}(\phi_\Theta(A, X)_{[c][i]}, \phi_\Theta(A, X)_{[c][j]}) \right], \\
& w_{i,j} = \mathrm{softmax}\left( \frac{Q_{i,j}^{\mathrm{diff}}}{d(i,j) Q_{i,j}^{\mathrm{same}}} \right),
\end{aligned}
\tag{13}
$$

where $\Omega_c(t) = \{(i,j) | i \neq j \wedge Y_i = Y_j = c \wedge d(v_i, v_j) \leq t\}$. $d(i,j)$ is the number of the hops of the shortest path from node $v_i$ to $v_j$, $Q_{i,j}^{\mathrm{diff}} = \sum_{c' \neq c} \left| |\mathcal{N}_{v_i}^{c'}| - |\mathcal{N}_{v_j}^{c'}| \right|$, $Q_{i,j}^{\mathrm{same}} = \left| |\mathcal{N}_{v_i}^{c}| - |\mathcal{N}_{v_j}^{c}| \right|$, $|\mathcal{N}_v^c|$ is the number of the neighbors of $v$ with class $c$. $t \in \mathbb{N}^+$ is a hyperparameter. Note that LoRe-CIA does not require ground-truth environment labels or environmental inference. More importantly, it does not divide the whole graph into multiple subgraphs as EERM does, which blocks the message passing from one environment to another. This is harmful for tasks like CBAS (Gui et al., 2022) that require global information to make predictions. LoRe-CIA is better equipped to handle such tasks (see Table 2 and 3).

In practice, we use LoRe-CIA as a regularization term added to the cross entropy loss with a weight $\lambda$ as a hyperparameter. The detailed training process is in Appendix D.

## 5 EXPERIMENT

### 5.1 EXPERIMENT SETUP

We run experiments on GOOD (Gui et al., 2022), a graph OOD benchmark. We reported the results on two types of OOD shifts: covariate shift and concept shift, which correspond to the cases that $p(X)$ and $p(Y|X)$ are shifted, respectively. The detailed experimental setup and hyperparameter settings are in Appendix B. Since GOOD provides environment labels, we can evaluate methods using environment labels like IRM, VREx, and CIA. But in practical scenarios, the environment partition is hard to obtain.

### 5.2 OOD GENERALIZATION RESULTS

Table 2: OOD test accuracy (%) on covariate shift. The best and second-best results are shown in **bold** and underlined, respectively. '*' marks the results adopted from Gui et al. (2022) since we got out of memory at runtime.

| Dataset | Arxiv | | Cora | | CBAS | WebKB | average |
|---|---|---|---|---|---|---|---|
| Domain | degree | time | degree | word | color | university | |
| ERM (Vapnik, 1999) | 58.92(0.14) | 70.98(0.20) | 55.78(0.52) | 64.76(0.30) | 78.57(2.02) | 16.14(1.35) | 57.52 |
| IRM (Arjovsky et al., 2020) | 58.93(0.17) | 70.86(0.12) | 55.77(0.66) | 64.81(0.33) | 78.57(1.17) | 13.75(4.91) | 57.12 |
| VREx (Krueger et al., 2021) | 58.75(0.16) | 69.80(0.21) | 55.97(0.53) | 64.43(0.38) | 79.05(1.78) | 17.72(11.27) | 57.62 |
| GroupDRO (Sagawa et al., 2019) | 58.87(0.00) | 70.93(0.09) | 55.64(0.50) | 64.62(0.30) | 79.52(0.67) | 14.29(2.59) | 57.31 |
| DANN (Ganin et al., 2016) | 59.03(0.15) | 71.09(0.03) | 55.84(0.58) | 64.74(0.32) | 80.95(1.78) | 16.67(1.29) | 58.05 |
| Deep Coral (Sun & Saenko, 2016) | 59.04(0.16) | 71.04(0.07) | 56.03(0.37) | 64.75(0.26) | 78.09(0.67) | 11.90(1.72) | 56.81 |
| EERM (Wu et al., 2021) | OOM | OOM | 56.88(0.32)* | 61.98(0.10)* | 40.48(9.78) | 16.21(5.67) | - |
| SRGNN (Zhu et al., 2021) | 58.47(0.00) | 70.83(0.10) | 57.13(0.25) | 64.50(0.35) | 73.81(4.71) | 16.40(1.63) | 56.86 |
| Mixup (Wang et al., 2021) | 57.80(0.19) | **71.62(0.11)** | **57.89(0.27)** | 65.07(0.22) | 70.00(5.34) | 16.67(1.12) | 56.51 |
| CIA (Mahajan et al., 2021) | 59.03(0.39) | 71.10(0.15) | 56.34(0.35) | 65.07(0.52) | 78.57(1.17) | 18.25(2.33) | 58.06 |
| LoRe-CIA (ours) | **59.12(0.18)** | 71.16(0.11) | 57.06(0.27) | **65.16(0.09)** | **81.43(0.00)** | 18.52(2.28) | **58.74** |

Table 3: OOD test accuracy (%) on concept shift.

| Dataset | Arxiv | | Cora | | CBAS | WebKB | average |
|---|---|---|---|---|---|---|---|
| Domain | degree | time | degree | word | color | university | |
| ERM (Vapnik, 1999) | 62.92(0.21) | 67.36(0.07) | 60.24(0.40) | 64.32(0.15) | 82.14(1.17) | 27.52(0.75) | 60.75 |
| IRM (Arjovsky et al., 2020) | 62.79(0.11) | 67.42(0.08) | 61.23(0.08) | 64.42(0.18) | 81.67(0.89) | 27.52(0.75) | 60.84 |
| VREx (Krueger et al., 2021) | 63.06(0.43) | 67.42(0.07) | 60.69(0.42) | 64.32(0.22) | 82.86(1.17) | 27.52(1.50) | 60.98 |
| GroupDRO (Sagawa et al., 2019) | 62.98(0.53) | 67.41(0.27) | 60.59(0.36) | 64.34(0.25) | 82.38(0.67) | 28.44(0.00) | 61.02 |
| DANN (Ganin et al., 2016) | 63.04(0.20) | 67.46(0.23) | 60.32(0.26) | 64.34(0.12) | 82.86(0.58) | 26.61(1.50) | 60.77 |
| Deep Coral (Sun & Saenko, 2016) | 63.09(0.28) | 67.43(0.24) | 60.41(0.27) | 64.34(0.17) | 82.86(0.58) | 26.61(0.75) | 60.79 |
| EERM (Wu et al., 2021) | OOM | OOM | 58.38(0.04)* | 63.09(0.36)* | 61.43(1.17) | 28.04(11.67) | - |
| SRGNN (Zhu et al., 2021) | 62.80(0.25) | 67.17(0.23) | 61.21(0.29) | 64.53(0.27) | 80.95(0.67) | 27.52(0.75) | 60.70 |
| Mixup (Wang et al., 2021) | 62.33(0.34) | 65.28(0.43) | **63.65(0.37)** | 64.45(0.12) | 65.48(0.67) | **30.28(1.50)** | 58.58 |
| CIA (Mahajan et al., 2021) | 63.87(0.26) | **67.62(0.04)** | 61.59(0.18) | 64.61(0.11) | 85.71(0.72) | 27.83(1.89) | 61.84 |
| LoRe-CIA (ours) | **63.89(0.31)** | 67.52(0.10) | 62.09(0.33) | **64.62(0.17)** | **85.72(1.54)** | 28.75(0.43) | **62.10** |

The results on covariate shift and concept shift datasets are in Table 2 and 3, respectively. Although the LoRe-CIA was not optimal on some datasets, it achieves average optimality and did not show results far below average on any dataset, which demonstrates its stability. The failure of EERM on CBAS stems from its subgraph partitioning process, which causes the node representations to lose global information, thereby unable to predict the node's position on the graph, which is precisely the task of CBAS. LoRe-CIA effectively addresses this issue by avoiding dividing multiple subgraphs.

CIA outperforms IRM and VREx on all datasets except that it fails to beat VREx on CBAS (covariate), which validates our findings in Section 3. LoRe-CIA outperforms CIA on all datasets, indicating our reweighting strategy can further benefit generalization.

Due to space limitation, we have placed the analysis of the impact of hyperparameters $\lambda$ and $t$ and the visualization of the LoRe-CIA representations in Appendix C.

# 6 RELATED WORK

## 6.1 INVARIANT LEARNING FOR OUT-OF-DISTRIBUTION GENERALIZATION

Invariant learning seeks to find stable features across multiple training environments to achieve OOD generalization. The goal of IRM (Arjovsky et al., 2020) is to learn a representation that elicits a classifier achieving optimality in all training environments. To solve the bi-leveled optimization problem, Arjovsky et al. (2020) proposed IRMv1 that minimizes the norm of the gradients w.r.t. the classifier in all training environments. REx (Krueger et al., 2021) reduces the risks across training environments to improve robustness against distribution shifts. They proposed two practical algorithms MMREx and VREx for risk extrapolation. In this paper, we consider the more widely used VREx. Mahajan et al. (2021) proposed MatchDG to match the representations of the same causal features across domains. To the best of our knowledge, unlike VREx which has been evaluated widely on graph OOD benchmarks, no previous work has investigated the effect of MatchDG (which we rename CIA in this paper) on graph OOD generalization (neither empirically nor theoretically, while we both do), and we are the first work to theoretically analysis the limitations of IRMv1 and VREx in OOD node classification problems.

## 6.2 NODE-LEVEL OUT-OF-DISTRIBUTION GENERALIZATION ON GRAPHS

There has been a substantial amount of work focusing on the OOD generalization problem on graphs. However, the vast majority have centered on graph classification tasks(Chen et al., 2022; Zhu et al., 2021; Wu et al., 2022; Li et al., 2022b; 2023; Yang et al., 2022; Yu et al., 2023; Chen et al., 2023; Buffelli et al., 2022) and only a small amount of work has focused on the node-level OOD task (Wu et al., 2021; Liu et al., 2023; Li et al., 2022a; Zhu et al., 2021). (Wu et al., 2021) proposed EERM, which first generates multiple training environments that maximize the variance of the risks and then applies VREx on the generated environments. However, it adopts an adversarial training manner for the environment generator, which is unstable and could induce suboptimal performances (Figure 2 and 3). SRGNN (Zhu et al., 2021) aims at aligning the representation of the biased training samples and the unlabeled i.i.d data. There may be some problems with this. First, it only aligns the marginal distribution $p(X)$, which has been proved to have failure cases (Johansson et al., 2019). Second, it does not restrict the aligned samples to have small distribution differences in causal features (like Mahajan et al. (2021) suggested). This may lead to a loss of diversity in causal features during alignment. Our proposed method fixes these issues: (1) LoRe-CIA avoids complex environment inference but can still create differences in spurious features; (2) LoRe-CIA considers the failure case caused by varying causal features by adding adaptive weights to the alignment terms.

# 7 CONCLUSION

In this work, we find the failure of IRMv1 and VREx at node-level graph OOD tasks and suggest that CIA is a more appropriate optimization target. To solve the problem of lack of environment labels and feature collapse, we propose LoRe-CIA, which further improves graph OOD generalization performance. However, how to make better use of graph structure to identify the pattern of spurious features and specific characteristics of distribution shifts on graphs is a problem to be further explored.

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

# A ADDITIONAL THEORETICAL RESULTS AND PROOFS OF THE COVARIATE SHIFT CASE

## A.1 THEORETICAL MODEL SETUP OF THE COVARIATE SHIFT CASE

In this section, we will extend our theoretical model in the main text to the covariate shift setting. For the covariate shift setting, spurious features are independent of $Y$. Thus we can model the data generation process for environment $e$ as

$$Y^e = \tilde{A}^{e^k} X_1 + n_1, \quad X_2^e = n_2 + \epsilon^e, \tag{14}$$

where the definition of $n_1$ and $n_2$ are the same as Section 3, $\epsilon^e$ represents environmental spurious features. $\epsilon_i^e$ (each dimension of $\epsilon^e$) is a random variable that are independent for $i = 1, ..., N^e$. We assume the intra-environment expectation of the environment spurious variable is $\mathbb{E}_{\epsilon_i \sim p_e}[\epsilon_i] = \mu^e \in \mathbb{R}$ since spurious features are consistent in a certain environment. We further assume the cross-environment expectation $\mathbb{E}_e[\epsilon^e] = \mathbf{0}$ and cross-environment variance $\mathbb{E}_e[\epsilon_i^e] = \sigma^2$, $i = 1, ..., N^e$ for simplicity. This is consistent with the covariate shift case that $p(X)$ can arbitrarily change across different domains, and the support set of $X$ may vary. Note that different from the concept shift setting, we only require $L \geq k$ to ensure the predictiveness of the network.

## A.2 THEORETICAL RESULTS OF THE COVARIATE SHIFT CASE

In this subsection, we will present the failure case of VREx and IRMv1, and the success case of CIA under covariate shift (which is a different setting from the results of the concept shift case in the main text). The proofs are in Appendix E.2.

### A.2.1 THE FAILURE CASE OF VREX UNDER COVARIATE SHIFT

**Proposition A.1.** *(VREx will use spurious features) The objective $\min_{\Theta} \mathbb{V}_e[R(e)]$ has non-unique solutions, and when part of the model parameters $\{\theta_1^{1^{(l)}}, \theta_1^{2^{(l)}}, \theta_2^{1^{(l)}}, \theta_2^{2^{(l)}}\}$ take the values*

$$\Theta_0 = \begin{cases} \theta_1^{1^{(l)}} = 1, \theta_1^{2^{(l)}} = 1, & l = L-1, ..., L-s+1 \\ \theta_1^{1^{(l)}} = 0, \theta_1^{2^{(l)}} = 1, & l = L-s, L-s-1, ..., 1 \\ \theta_2^{1^{(l)}} = 0, \theta_2^{2^{(l)}} = 1, & l = L-1, ..., 1 \end{cases}, \tag{15}$$

*$0 < s < L$ is some positive integer, $\theta_1$ and $\theta_2$ have four sets of solutions of the quadratic equation:*

$$\begin{cases} c_1 \sigma^2 (2\theta_1 \theta_2 + (\theta_2)^2 - 2c_2 \sigma^2 \theta_2) + c_3 - \mathbb{E}_e[N^e] c_1 \sigma^2 \theta_1 \theta_2 + \mathbb{E}_e[N^e] c_2 \sigma^2 \theta_2 = 0 \\ [c_3 - \mathbb{E}_e[N^e] c_1 \sigma^2 \theta_1 \theta_2 + \mathbb{E}_e[N^e] c_2 \sigma^2 \theta_2] c_4 - c_5 (\theta_2)^2 = 0 \end{cases}. \tag{16}$$

*where* $c_1 = \mathbb{E}[(\tilde{A}^{e^s} X_1)^\top (\tilde{A}^{e^s} X_1)]$, $c_2 = \mathbb{E}[(\tilde{A}^{e^s} X_1)^\top (\tilde{A}^{e^k} X_1)]$, $c_3 = \mathbb{E}_e[\epsilon^{e\top} \epsilon^e \epsilon^{e\top} (\tilde{A}^{e^s} X_1)]\sigma^2, c_4 = \mathbb{E}_e\left[(\tilde{A}^{e^k} X_1)^\top \mathbf{1}_{N^e}\right]\sigma^2$, $c_5 = \mathbb{E}_e\left[N^e\left(tr((\tilde{A}^{e^k})^\top \tilde{A}^{e^k}) + N^e(1+\sigma^2)\right)\right]$.

**Remark.** For the covariate shift setting, $\theta_2 = 0$ is still not a solution to the VREx objective in node-level OOD tasks. Therefore it will also rely on spurious features.

### A.2.2 THE FAILURE CASE OF IRMV1 UNDER COVARIATE SHIFT

**Proposition A.2.** *(IRMv1 will use spurious features) The objective $\min_{\Theta} \mathbb{E}_e[\|\nabla_{w|w=1.0} R(e)\|^2]$ has a solution that the invariant parameter $\theta_1$ will produce inaccurate predictions,*

$$\theta_1 = \frac{\mathbb{E}_e[(\tilde{A}^{e^k} X_1)^\top (\tilde{A}^{e^{2k}} X_1)]}{\mathbb{E}_e[(\tilde{A}^{e^{2k}} X_1)^\top (\tilde{A}^{e^{2k}} X_1)]} \tag{17}$$

*and there will be no constraints on the spurious parameter $\theta_2$, when $\{\theta_1^{1(l)}, \theta_1^{2(l)}, \theta_2^{1(l)}, \theta_2^{2(l)}\}$ take the special values for some $0 < s < L$:*

$$\Theta_0 = \begin{cases} \theta_1^{1(l)} = 1, \theta_1^{2(l)} = 1, & l = L-1, ..., L-s+1 \\ \theta_1^{1(l)} = 0, \theta_1^{2(l)} = 1, & l = L-s, L-s-1, ..., 1 \\ \theta_2^{1(l)} = 0, \theta_2^{2(l)} = 1, & l = L-1, ..., 1 \end{cases} \quad . \tag{18}$$

### A.2.3 THE SUCCESSFUL CASE OF CIA UNDER COVARIATE SHIFT

**Proposition A.3.** *Optimizing the CIA objective will lead to the optimal solution $\Theta^*$:*

$$\begin{cases} \theta_1 = 1 \\ \theta_2 = 0 \\ \theta_1^{1(l)} = 1, \theta_1^{2(l)} = 1, & l = L-1, ..., L-k+1 \\ \theta_1^{1(l)} = 0, \theta_1^{2(l)} = 1, & l = L-k, L-k-1, ..., 1 \end{cases} \quad . \tag{19}$$

## B DETAILED EXPERIMENTAL SETUP

### B.1 BASIC SETTINGS

All experimental results were averaged over three random runs. Following (Gui et al., 2022), we use an OOD validation set for model selection and use a 3-layer GCN (Kipf & Welling, 2016) as the backbone GNN, except that Mixup uses a modified GCN. The settings for learning rate, batch size, and training epochs also follow (Gui et al., 2022).

### B.2 HYPERPARAMETER SETTINGS

Most hyperparameter settings are adopted from (Gui et al., 2022), except that for EERM we reduce the number of generated environments from 10 to 7 and reduce the number of adversarial steps from 5 to 1 for memory and computing complexity concerns. For each parameter of the methods, we conduct a grid search for about 3~4 values.

## C ADDITIONAL EXPERIMENTAL RESULTS

### C.1 PARAMETER ANALYSIS

In this section, we analyze the effect of $\lambda$ and the number of adjacent hops of LoRe-CIA. From Figure 2, we can see clearly that adding LoRe-CIA regularization is beneficial for generalization since the test accuracy increases with $\lambda$. Note that most of the parameter combinations outperform the baseline methods (ERM: 55.78/60.24, IRM: 55.77/61.23, VREx: 55.97/60.69), indicating that our method leads to consistently superior performance.

On Cora degree covariate shift, we can observe the positive effect of localized alignment: with the decrease of $t$, the accuracy rate increases gradually. However, the trend is not clear in the covariate shift. On covariate shift, the accuracy rate varies differently with $t$ for different $\lambda$, indicating that there is a synergistic effect on the accuracy rate. How to better balance these two parameters is a direction worth exploring in the future.

### C.2 REPRESENTATION VISUALIZATION

We visualize the representation of CIA and LoRe-CIA in Figure 3 to show that LoRe-CIA can alleviate feature collapse caused by overalignment.

Table 4: Hyperparameter setting of the experiments.

| Algorithm | Search Space |
|---|---|
| IRM | 0.1, 1, 10, 100 |
| VREx | 1, 10, 100, 1000 |
| GroupDRO | 0.001, 0.01, 0.1 |
| DANN | 0.001, 0.01, 0.1 |
| Deep Coral | 0.01, 0.1, 1 |
| Mixup | 0.4, 1.0, 2.0 |
| EERM | $\beta$=0.5, 1, 3
number of generated environments $k$=7
adversarial training steps $t$=1
numbers of nodes for each node should be modified the link with $s$=5
subgraph generator learning rate $r$=0.0001, 0.001, 0.005, 0.01 |
| SRGNN | 0.000001, 0.00001, 0.0001 |
| CIA | $\lambda$=0.0001, 0.001, 0.005, 0.01, 0.05, 0.1 |
| LoRe-CIA | $\lambda$= 0.001, 0.005, 0.01, 0.05, 0.1
hops $t$=2, 3, 4, 5 |

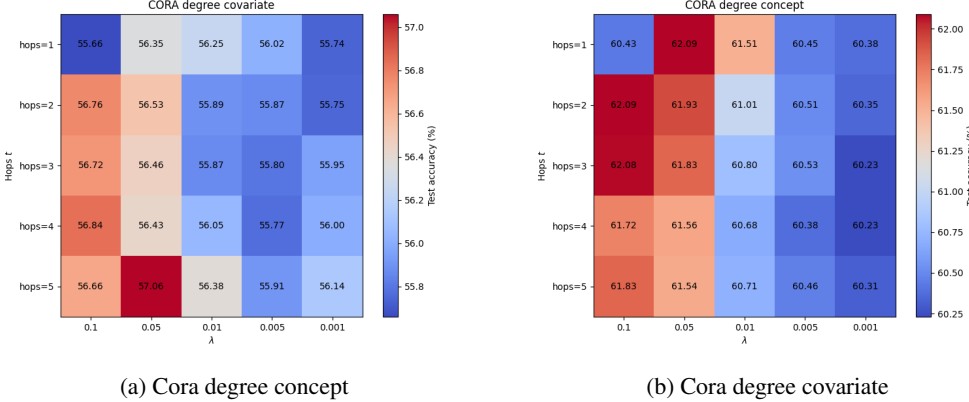

(a) Cora degree concept        (b) Cora degree covariate

Figure 2: Effect of $\lambda$ and the number of hops on OOD test accuracy (%).

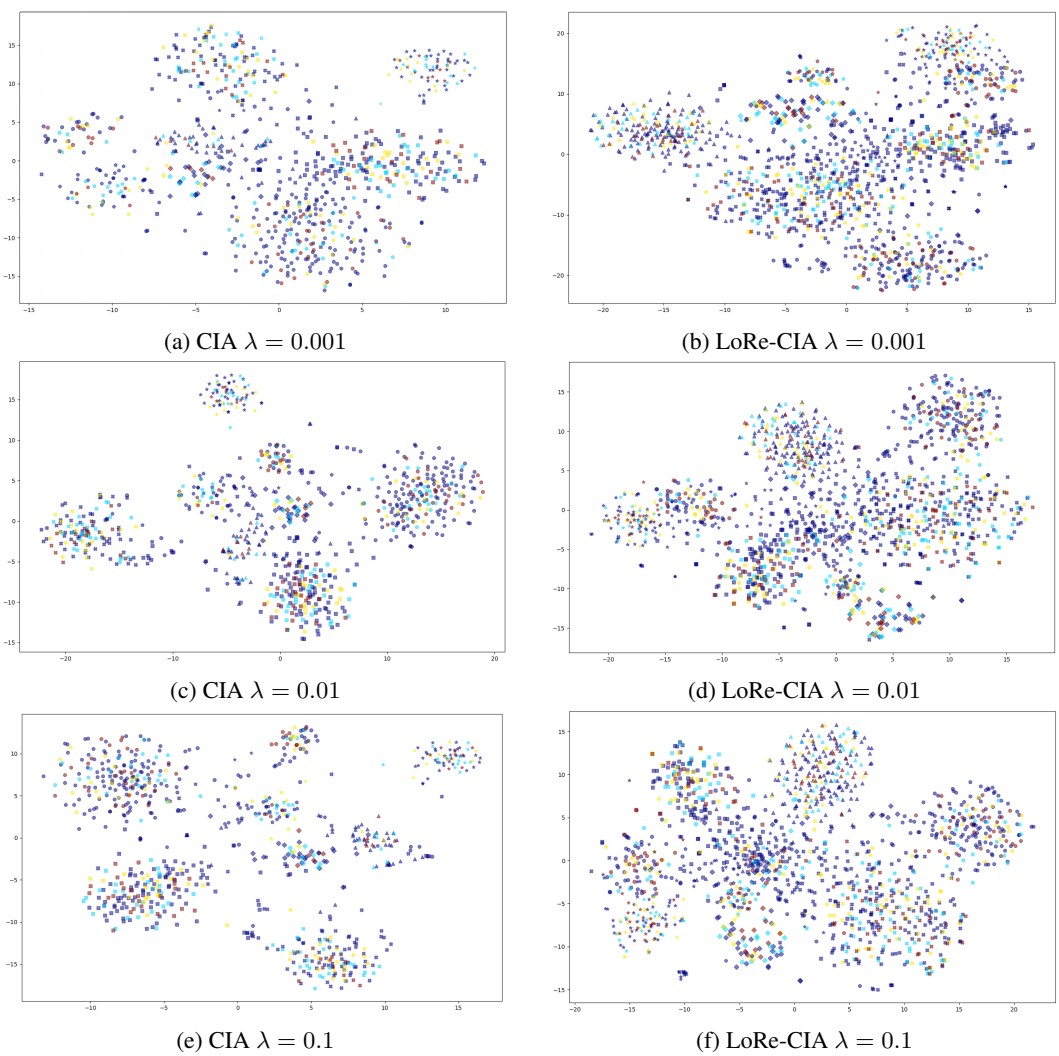

Figure 3: Visualization of the learned representations of nodes. LoRe-CIA can prevent the features of each class from being too concentrated.

### C.3   Validation of the True Feature Generation Depth

For the theoretical model in section 3, we assume $L \geq k$. To empirically find out how large $k$ really is, we use GCN with different layer to predict the ground-truth label $Y$ on Cora and Arxiv dataset respectively (results are in Table 5 and 6). As mentioned above, since a GCN with layer $l$ will aggregate features from $l$-hop neighbors for prediction, if the depth of the GCN is equal to the true generation depth, then the performance should be close to optimal. Suppose the empirical optimal layer number is $L^*$ for prediction, we have: $L^* = k$ **We find that the $L^*_s \leq 4$ in most cases (even on large-scale graphs in Arxiv).** This indicates that our assumptions holds easily.

Table 5: OOD accuracy (%) of GCN with different numbers of layers on Cora.

| Dataset | Shift | $L = 1$ | $l = 2$ | $L = 3$ | $L = 4$ |
|---|---|---|---|---|---|
| Cora (degree) | covariate | **59.04(0.15)** | 58.44(0.44) | 55.78(0.52) | 55.15(0.24) |
| | concept | **62.88(0.34)** | 61.53(0.48) | 60.24(0.40) | 60.51(0.17) |
| Cora (word) | covariate | 64.05(0.18) | **65.81(0.12)** | 65.07(0.52) | 64.58(0.10) |
| | concept | 64.76(0.91) | **64.85(0.10)** | 64.61(0.11) | 64.16(0.23) |

Table 6: OOD accuracy on causal prediction (%) of GCN with different numbers of layers on Arxiv.

| Dataset | Shift | $l = 2$ | $L = 3$ | $L = 4$ | L=5 |
|---|---|---|---|---|---|
| Arxiv (degree) | covariate | 57.28(0.09) | 58.92(0.14) | **60.18(0.41)** | 60.17(0.12) |
| | concept | 63.32(0.19) | 62.92(0.21) | **65.41(0.13)** | 63.93(0.58) |
| Arxiv (time) | covariate | 71.17(0.21) | 70.98(0.20) | **71.71(0.21)** | 70.84(0.11) |
| | concept | 65.14(0.12) | 67.36(0.07) | 65.20(0.26) | **67.49(0.05)** |

### C.4   Discussion and Validation of the Assumption on the Rate of Change of Causal and Spurious Features w.r.t Spatial Position

To verify the intuition used in Section 4.2 that the change rate of node's spurious features w.r.t spatial location is faster than that of the causal/invariant features within a certain range of hops, we conduct experiments on GOOD-Arxiv and GOOD-Cora, both are real-world citation networks. To extract invariant features, we use a pretrained VREx model and take the output of the last layer as invariant features[3]. To obtain spurious features, we train a ERM model to predict the environment label and also take the output of the last layer as spurious features. For each class, we randomly sample 10 nodes and generate corresponding 10 paths using Breadth-First Search (BFS). We extract invariant and spurious features of the nodes on each paths, and plot the distances between the node representations on the paths and the starting node. The results of Cora are in Figure 4 and 5, and the results of Arxiv are in Figure 6 and 7. (we choose some of the classes to avoid excessive paper length, the results for the other classes are similar).

We can see that within about 5∼10 hops, the changes of spurious features grow more rapidly than invariant ones. Hence we propose to align the representations of adjacent nodes to better eliminate spurious features and avoid the collapse of the invariant features. And this explains we add a weighting term $d(i, j)$ in our loss function to assign smaller weight node pairs farther apart.

This assumption is similar to the ones adopted by a series of previous works on causality and invariant learning (Chen et al., 2022; Burshtein et al., 1992; Schölkopf, 2022; Schölkopf et al., 2021).

---

[3]though we reveal in our theory that VREx could rely on spurious features, we still use VREx here to approximately extract invariant features as many previous graph OOD works did since VREx already gains some advantages.

They assume causal features are more well-clustered than spurious features. In node-level graph OOD scenario, we observe this phenomenon only within local parts of a graph. In some cases, when two nodes are too far away from each other, their causal features can also vary more than the spurious features, as can been seen in Figure 7 (a) path 1,2,4,6,9 and 10. Therefore, choosing to match the representations in a local region can help to alleviate the feature collapse problem.

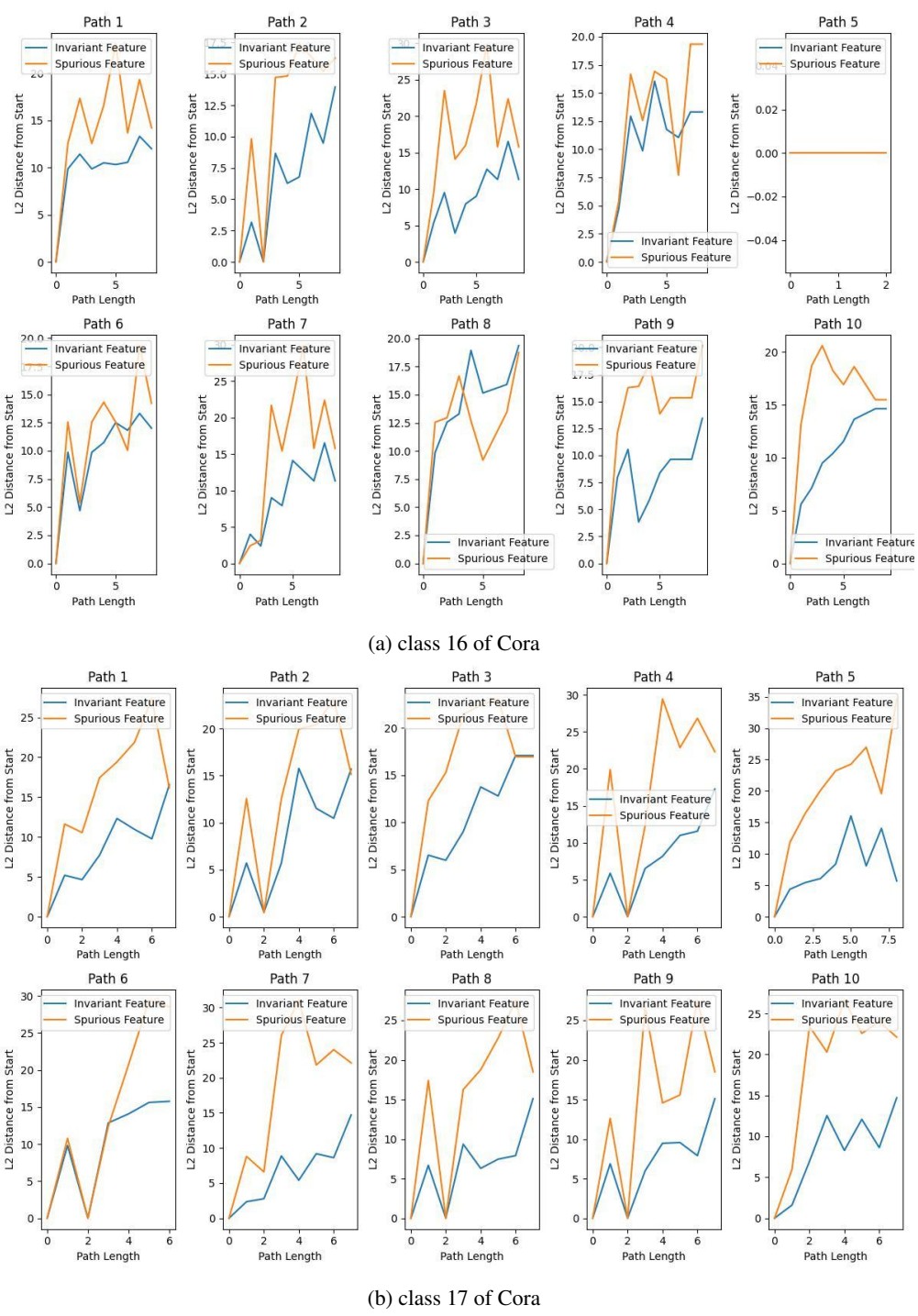

(a) class 16 of Cora

(b) class 17 of Cora

Figure 4: Visualization of the rate of change of invariant features and spurious features on Cora (part 1).

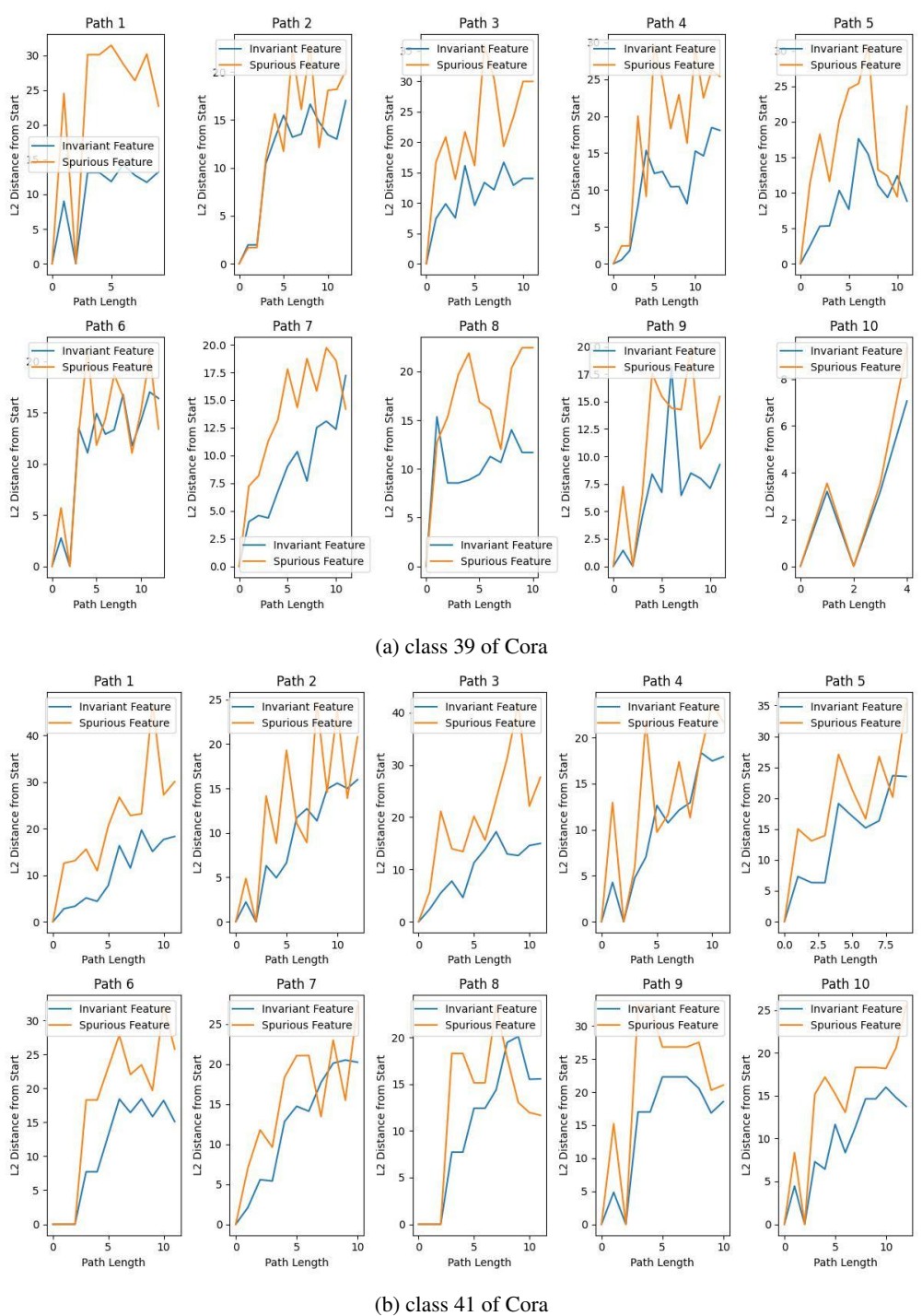

Figure 5: Visualization of the rate of change of invariant features and spurious features on Cora (part 2).

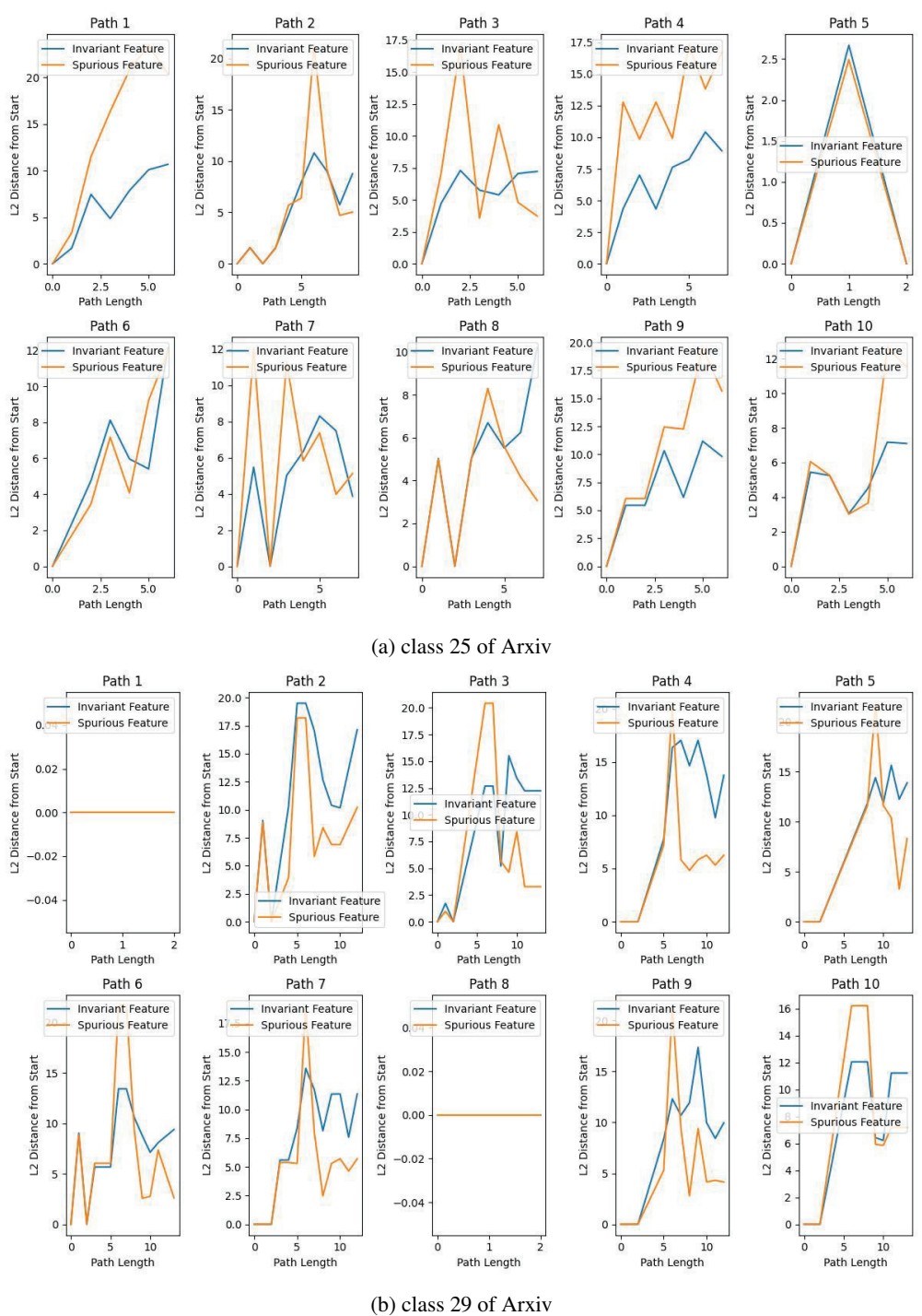

(a) class 25 of Arxiv

(b) class 29 of Arxiv

Figure 6: Visualization of the rate of change of invariant features and spurious features on Arxiv (part 1).

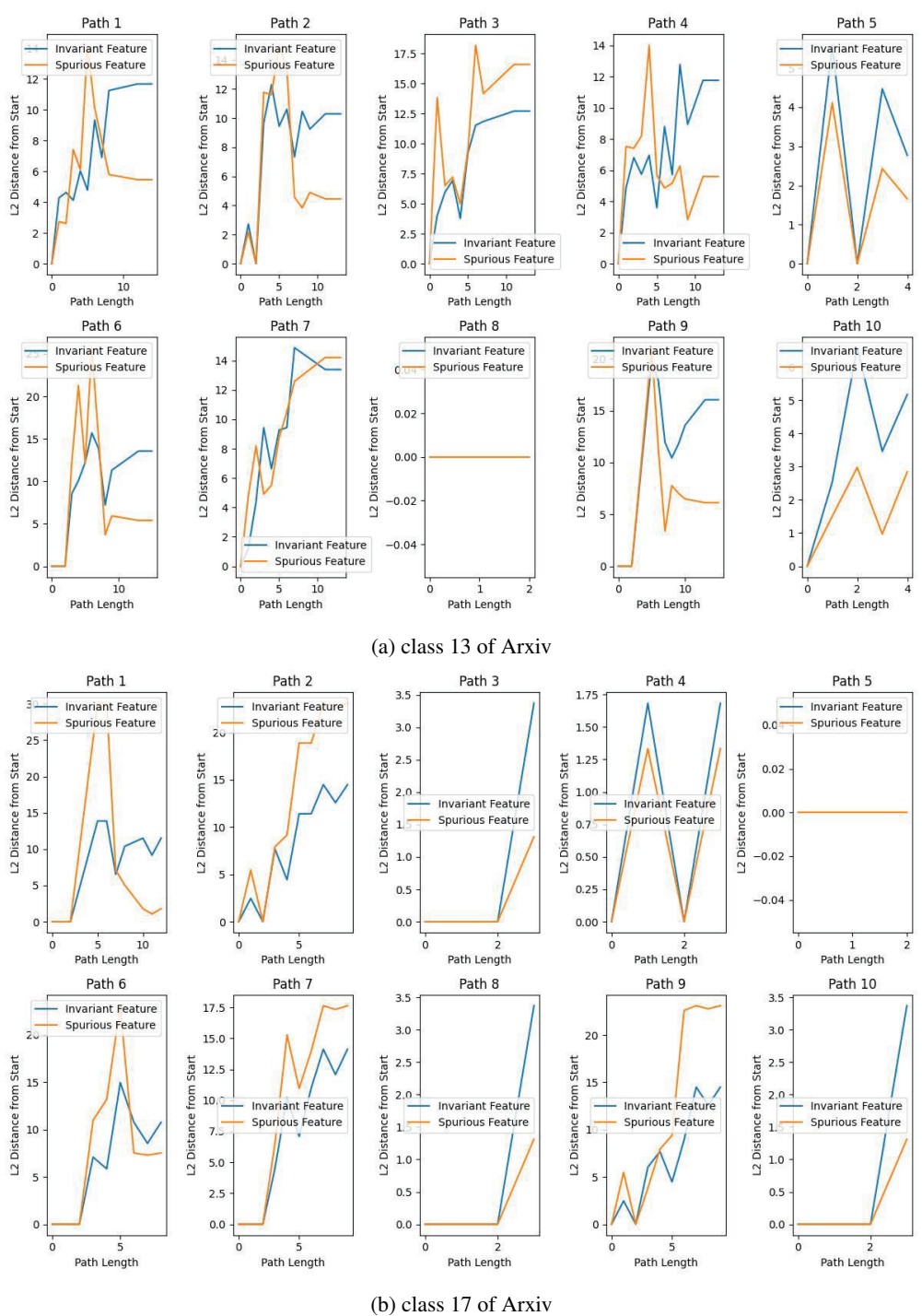

(a) class 13 of Arxiv

(b) class 17 of Arxiv

Figure 7: Visualization of the rate of change of invariant features and spurious features on Arxiv (part 2).

## C.5 Discussion and Validation of the Assumption on the Feature Distance and Neighboring Label Distribution Discrepancy

### C.5.1 Class-different Neighboring Labels Reflect Spurious Feature Distribution

In this section, we will empirically validate the key intuition of LoRe-CIA: the label distribution of the neighbors from different classes (which we call *Heterophilous Neighboring Label Distribution (HNLD)* in the following contents) reflects the spurious feature of the centered node. This idea is similar to the observation in Song & Wang (2022): heterophily is a main source of node distribution shift. Moreover, as recommended by Ye et al. (2022), we will further investigate the impact of HNLD on spurious feature distribution under two types of OOD shift: *concept shift* (or correlation shift in (Ye et al., 2022)), where $p(Y|X)$ varies across environments, and *covariate shift* (or diversity shift in (Ye et al., 2022)), where $p(X)$ changes with environments, respectively. We will show that HNLD affect the spurious features of the centered node in different manners under concept shift and covariate shift.

Spurious features represent features that have no predictive power for labels, and spurious features of a node come from two sources: (1) the environmental spurious feature, i.e. features determined by environments that contain no invariant and predictive information about labels, (2) class-different (heterophilous) neighboring features. The first source of spurious features is mentioned all the time in OOD and Domain Generalization (DG) topics, and many recent works have revealed that heterophilous neighbors harm node classification performance (Ma et al., 2021; Huang et al., 2023). In the follow part, we will first point out how to approximately measure spurious features for covariate and concept shift, and empirically validate our intuition.

**Covariate shift.** For covariate shifts on graphs, since spurious features are not necessarily correlated with labels, the environmental spurious features cannot be reflected by HNLD. However, we can still measure the distribution of the spurious features caused by heterophilous neighbors. To extract spurious features induced by class-different labels, we train a 1-layer GCN that aggregates neighboring features and discards the features of the centered node. The reason why we use features from all neighbors rather than only heterophilous neighbors is we want to simulate message-passing as authentically as possible, that is, we hope to observe whether the gap of HNLD accurately reflects the distance of heterophilous neighboring feature in the presence of both homophilous and heterophilous neighbors. To ensure that the discrepancy in the aggregated neighboring feature is caused solely by heterophilous neighbors, we only use point pairs with the same number of homophilous neighbors. Specifically, we compute the L2 distance between the neighbor representations of two nodes with the same number of class-same neighbors, and plot its trend w.r.t. the distance of HNLD (according to the definition of $Q_{i,j}^{\mathrm{diff}}$ in Equation 13). We run experiments on Cora to verify this. We evaluate on both *word* shifts (node feature shifts) and *degree* (graph structure shifts) for a comprehensive understanding. We show the results of first 30 classes of Cora. **The results in Figure 8 and 9 show a clear positive correlation between the spurious feature distance and HNLD discrepancy under covariate shifts.**

**Concept shift.** As for concept shift, spurious features are correlated with labels, thus the label of a node contains information about spurious features correlated with this class. Moreover, due to the massage-passing mechanism of GNNs, the spurious features of a centered node are also affected by neighboring nodes. Assuming that most adjacent nodes are from the same environment, the spurious features of same-class neighbors will not change that of the centered node since the spurious distribution is fixed given the class and the environment (Yi et al., 2022). Hence, by observing HNLD, we can measure the distribution of the spurious feature. For concept shift, we train a GNN to predict environment labels to obatin spurious representations. **Table 10 and 11 also show a clear positive correlation between spurious featured distance and HNLD discrepancy on concept shift.**

### C.5.2 Class-same Neighboring Labels Reflect Invariant Feature Distribution

Now will validate that the label distribution of the neighbors from the same class as the centered node reflects the invariant feature of the centered node. We use VREx to approximate invariant features, and compute the their distance w.r.t. the discrepancies of the neighboring label distribution of the same class. We evaluate on 4 splits of Cora: *word*+**covariate**, *word*+**concept**, *degree*+**covariate**

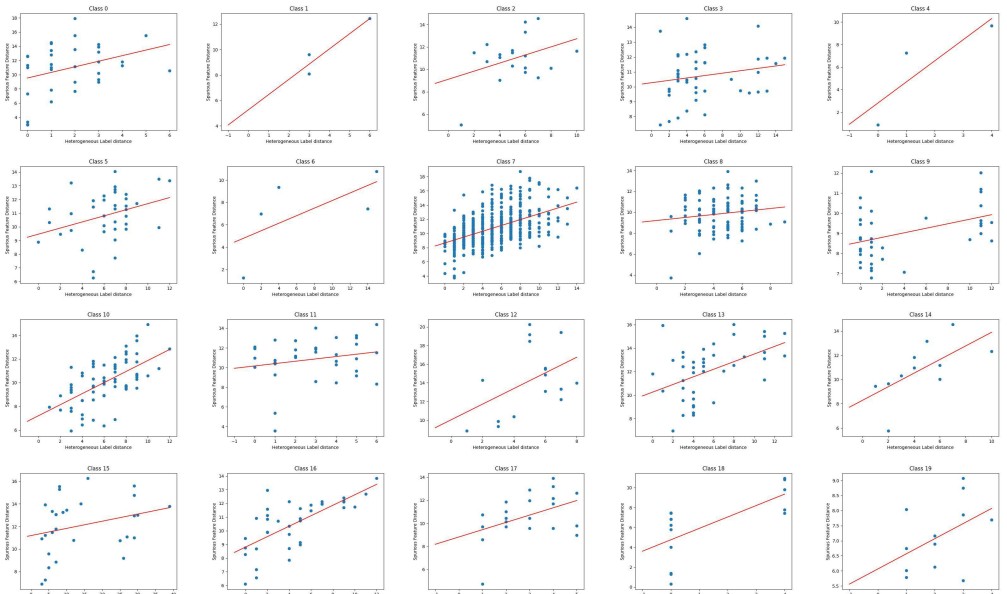

Figure 8: The relationship between the distance of spurious features induced by class-different neighbors and distance of HNLD on Cora *word* domain, **covariate shift**. Each sub-figure is a class, and each dot in the figure represents a node pair in the graph. The red line is obtained by linear regression. The positive correlation is clear.

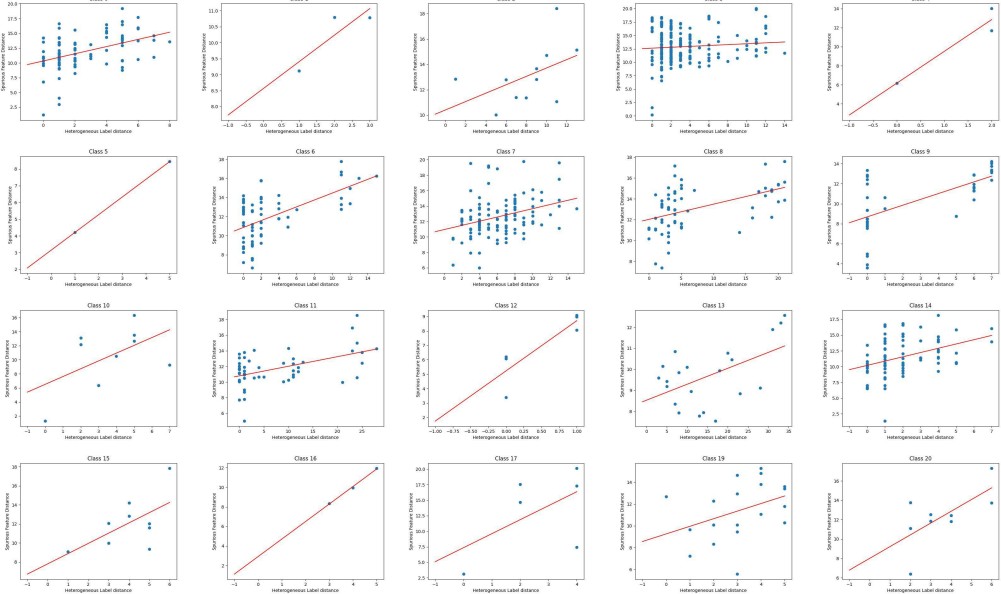

Figure 9: The relationship between the distance of spurious features induced by class-different neighbors and distance of HNLD on Cora *degree* domain, **covariate shift**. The positive correlation is clear.

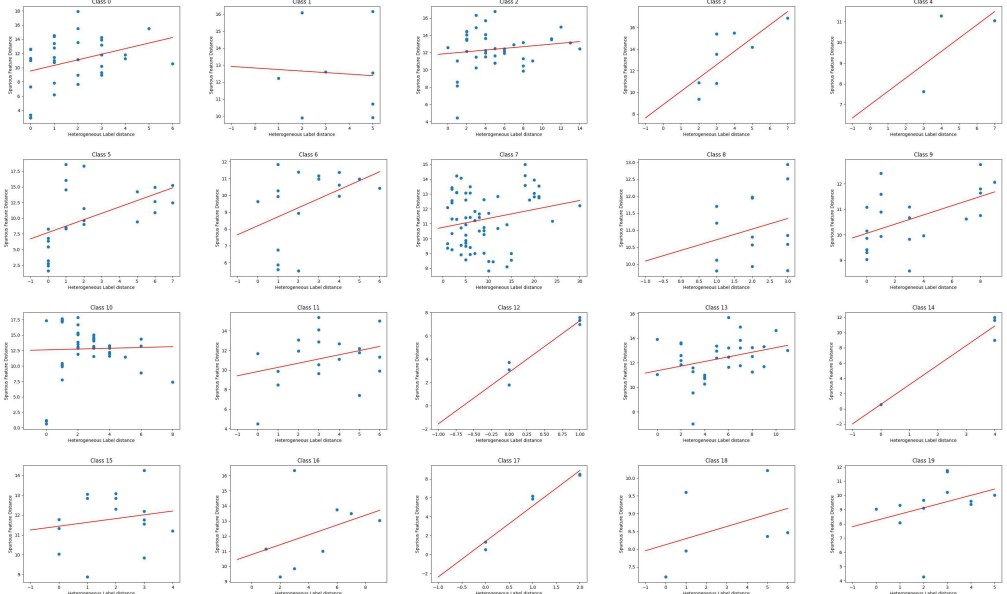

Figure 10: The relationship between the distance of environmental spurious features and distance of HNLD on Cora *word*, **concept shift**. The positive correlation is clear.

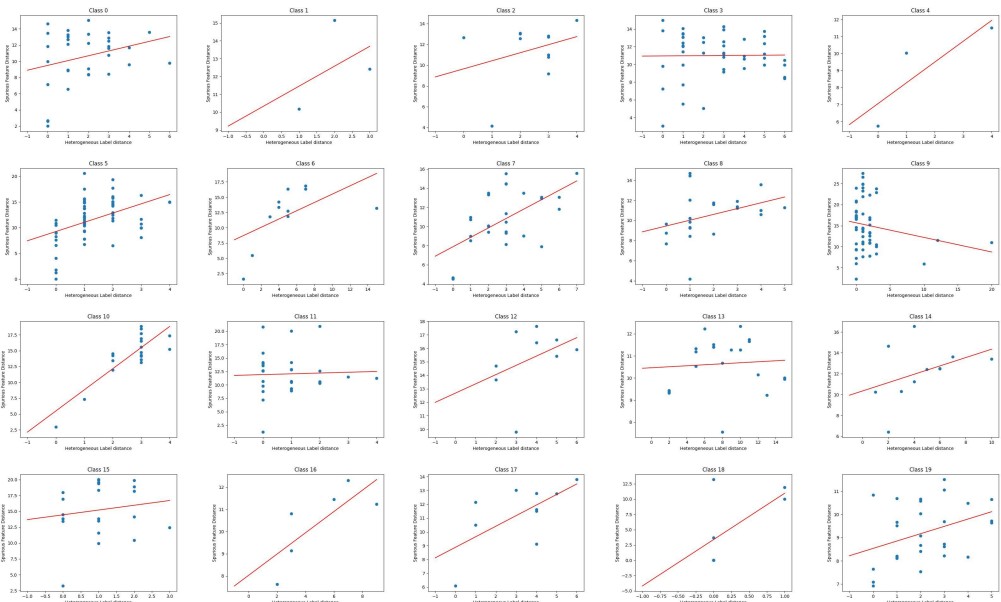

Figure 11: The relationship between the distance of environmental spurious features and distance of HNLD on Cora *degree*, **concept shift**. The positive correlation is clear.

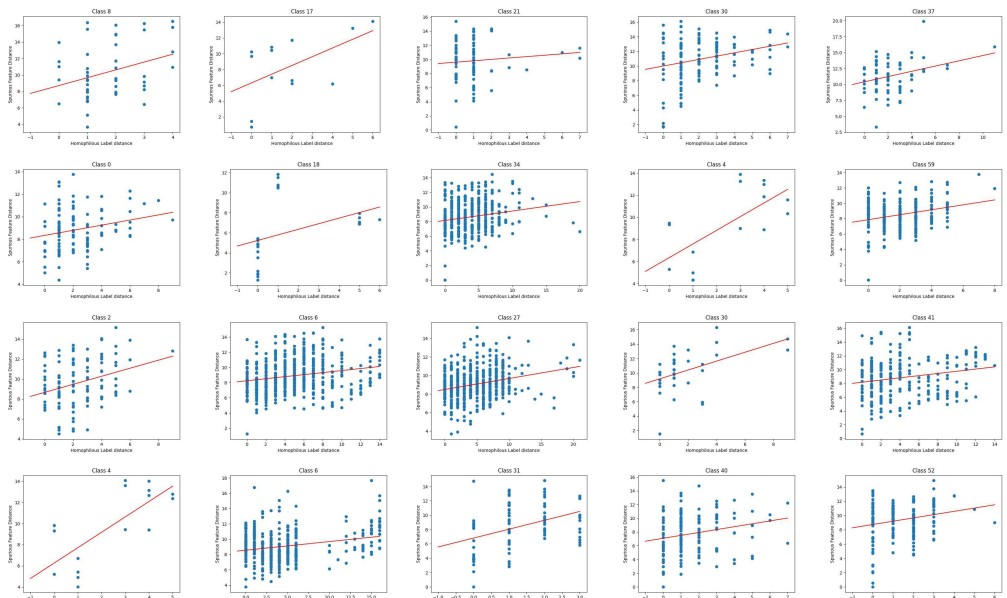

Figure 12: The relationship between the distance of invariant features and discrepancy of class-same neighboring labels on Cora *degree*, **concept shift**. Line 1 to 4 are results of Cora *word*+**covariate**, *word*+**concept**, *degree*+**covariate** and *degree*+**concept**, respectively. There is a positive correlation between the invariant feature distance and difference in neighboring labels of the same class as the centered node.

and *degree*+**concept**. For each data split, we randomly choose 5 classes that have node pairs with difference of larger than 5 in class-same neighboring labels. **The results in Table 12 also show a positive correlation trend.**

## D    DETAILED TRAINING PROCESS

Table 1 show the detailed training process of LoRe-CIA.

## E    PROOFS

### E.1    PROOFS OF THE CONCEPT SHIFT CASE PRESENTED IN THE MAIN TEXT

In this section, we give proof of the propositions of the concept shift model presented in the main text.

#### E.1.1    PROOF OF THE FAILURE CASE OF VREX UNDER CONCEPT SHIFT

**Proposition E.1.** *(VREx will use spurious features) The objective* $\min_\Theta \mathbb{V}_e[R(e)]$ *has non-unique solutions, and when part of the model parameters* $\{\theta_1^{1\,(l)}, \theta_1^{2\,(l)}, \theta_2^{1\,(l)}, \theta_2^{2\,(l)}\}$ *take the values*

$$\Theta_0 = \begin{cases} \theta_1^{1\,(l)} = 1, \theta_1^{2\,(l)} = 1, & l = L-1, ..., L-s+1 \\ \theta_1^{1\,(l)} = 0, \theta_1^{2\,(l)} = 1, & l = L-s, L-s-1, ..., 1 \\ \theta_2^{1\,(l)} = 0, \theta_2^{2\,(l)} = 1, & l = L-1, ..., 1 \end{cases}, \tag{20}$$

*for some* $0 < s < L$, $\theta_1$ *and* $\theta_2$ *have four sets of solutions of the cubic equation:*

$$\begin{cases} (3c_1\theta_1\theta_2 + c_1(\theta_2)^2 - 2c_6\theta_2)\sigma^2 - \mathbb{E}_e[N^e(2c_1(\theta_1 + \theta_2) - c_6)]\sigma^2\theta_2 + c_7 = 0 \\ (\mathbb{E}_e[N^e(2c_1(\theta_1 + \theta_2) - c_6)]\sigma^2\theta_2 - c_7)(c_3\theta_2 - c_4) - [c2(\theta_1 + \theta_2) - c_5](\theta_2)^2 = 0 \end{cases}. \tag{21}$$

---

**Algorithm 1** Detailed Training Procedure of LoRe-CIA

---

**Require:**
    A labeled training graph $\mathcal{G} = (A, X, Y)$.
    The number of hops $t$, LoRe-CIA weight $\lambda$, the number of classes $C$, total iterations $T$, model learning rate $r$.
**Ensure:**
    Updated model $f_\Theta$ with parameter $\Theta$.
 1: **for** iterations in $1, 2, ..., T$ **do**
 2:    Initialize $\mathcal{L}_{\text{LoRe}} = 0$
 3:    **for** $c$ in $1, 2, ..., C$ **do**
 4:        Calculate the node representations $\phi(A, X)$
 5:        Calculate $A_c^t$, where the $(i, j)$-th element of $A_c^t$ equals the length of the shortest path from node $i$ to $j$ if the length is less than $t$ else infinity.
 6:        Use $A_c^t$ to screen for pairs of nodes not exceeding a distance of $t$ hops $\Omega_c(t)$.
 7:        Compute LoRe-CIA loss of class $c$: $\mathcal{L}_{\text{LoRe}}^c$ according to Equation (13) using $\Omega_c(t)$, $A_c^t$ and $\phi(A, X)$.
 8:        $\mathcal{L}_{\text{LoRe}} = \mathcal{L}_{\text{LoRe}} + \mathcal{L}_{\text{LoRe}}^c$
 9:    **end for**
10:    Compute final loss $\mathcal{L} = \mathcal{L}_{\text{ce}}(f_\Theta(A, X), Y) + \lambda \mathcal{L}_{\text{LoRe}}$, $\mathcal{L}_{\text{ce}}$ is the cross entropy loss.
11:    Update model parameters $\Theta = \Theta - r\nabla_\Theta \mathcal{L}$
12: **end for**

---

*where* $c_1 = \mathbb{E}_e[(\tilde{A}^{e^s}X_1)^\top(\tilde{A}^{e^s}X_1)]$, $c_2 = \mathbb{E}_e[N_e(\tilde{A}^{e^s}X_1)^\top\tilde{A}^{e^s}X_1]$, $c_3 = \mathbb{E}_e[(\tilde{A}^{e^s}X_1)^\top\mathbf{1}]$, $c_4 = \mathbb{E}_e[((\tilde{A}^{e^k}X_1)^\top\mathbf{1}]$, $c_5 = \mathbb{E}_e[N_e((\tilde{A}^{e^k}X_1)^\top\tilde{A}^{e^s}X_1 + tr((\tilde{A}^{e^k})^\top\tilde{A}^{e^k}) + N^e(1 + \sigma^2))]$, $c_6 = \mathbb{E}_e[(\tilde{A}^{e^s}X_1)^\top(\tilde{A}^{e^k}X_1)]$, $c_7 = \mathbb{E}_e[\epsilon^{e\top}\epsilon^e\epsilon^{e\top}(\tilde{A}^{e^s}X_1)]$.

*Proof.* We will use some symbols to simplify the expression of the toy GNN. Denote $\tilde{A}^m n_1 + n_2 + \epsilon^e$ as $\eta$. Use the following notations to represent the components of the $L$-layer GNN model:

$$
\begin{aligned}
f_\Theta(A, X) &= H_1^{(L)}\theta_1 + H_2^{(L)}\theta_2 \\
&= \underbrace{\left[\theta_1^{1(L-1)}\bar{A}\left(...\theta_1^{1(3)}\left(\theta_1^{1(2)}\bar{A}(\theta_1^{1(1)}\bar{A} + \theta_1^{2(1)}\bar{I})X_1 + \theta_1^{2(2)}(\theta_1^{1(1)}\bar{A} + \theta_1^{2(1)}\bar{I})X_1\right) + ...\right)\right]}_{C_1}\theta_1 \\
&+ \underbrace{\left[\theta_2^{1(L-1)}\bar{A}\left(...\theta_2^{1(3)}\left(\theta_2^{1(2)}\bar{A}(\theta_2^{1(1)}\bar{A} + \theta_2^{2(1)}\bar{I})\tilde{A}^s X_1 + \theta_2^{2(2)}(\theta_2^{1(1)}\bar{A} + \theta_2^{2(1)}\bar{I})\tilde{A}^s X_1\right) + ...\right)\right]}_{C_2}\theta_2 \\
&+ \underbrace{\left[\theta_2^{1(L-1)}\bar{A}\left(...\theta_2^{1(3)}\left(\theta_2^{1(2)}\bar{A}(\theta_2^{1(1)}\bar{A} + \theta_2^{2(1)}\bar{I})\eta + \theta_2^{2(2)}(\theta_2^{1(1)}\bar{A} + \theta_2^{2(1)}\bar{I})\eta\right) + ...\right)\right]}_{Z}\theta_2 \\
&= C_1\theta_1 + (C_2 + Z)\theta_2.
\end{aligned}
\tag{22}
$$

$C_1, C_2, Z \in \mathbb{R}^{N \times 1}$. We use $C_1^e$, $C_2^e$, and $Z^e$ to denote the variables from the corresponding environment $e$. We further denote $C_2^e = C_2^{e\prime}\tilde{A}^{e^s}X_1$, $Z^e = C_2^{e\prime}\eta$.

Using these notations, the loss of environment $e$ is

$$
\begin{aligned}
R(e) &= \mathbb{E}_{n_1, n_2}\left[\|f_\Theta(A^e, X^e) - Y^e\|_2^2\right] \\
&= \mathbb{E}_{n_1, n_2}\left[\left\|C_1^e\theta_1 + (C_2^e + Z^e)\theta_2 - \tilde{A}^{e^k}X_1 - n_1\right\|_2^2\right].
\end{aligned}
\tag{23}
$$

Denote the inner term $C_1^e\theta_1 + (C_2^e + Z^e)\theta_2 - \tilde{A}^{e^k}X_1 - n_1$ as $l_e$.

The variance of loss across environments is:

$$
\begin{aligned}
\mathbb{V}_e[R(e)] &= \mathbb{E}_e[R^2(e)] - \mathbb{E}_e^2[R(e)] \\
&= \mathbb{E}_e\left[\left(\mathbb{E}_{n_1,n_2}\left\|C_1^e\theta_1 + (C_2^e + Z^e)\theta_2 - \tilde{A}^{e^k}X_1 - n_1\right\|_2^2\right)^2\right] \\
&\quad - \mathbb{E}_e^2\left[\mathbb{E}_{n_1,n_2}\left\|C_1^e\theta_1 + (C_2^e + Z^e)\theta_2 - \tilde{A}^{e^k}X_1 - n_1\right\|_2^2\right]. \\
&= \mathbb{E}_e\left[\mathbb{E}_{n_1,n_2}\left[(l_e^\top l_e)^2\right]\right] - \mathbb{E}_e^2\left[\mathbb{E}_{n_1,n_2}\left[l_e^\top l_e\right]\right].
\end{aligned}
\tag{24}
$$

Take the derivative of $\mathbb{V}_e[R(e)]$ with respect to $\theta_1$:

$$
\begin{aligned}
\frac{\partial \mathbb{V}_e[R(e)]}{\partial \theta_1} &= \mathbb{E}_e\left[2\mathbb{E}_{n_1,n_2}\left[l_e^\top l_e\right]\mathbb{E}_{n_1,n_2}\left[2l_e^\top C_1^e\right]\right] \\
&\quad - 2\mathbb{E}_e\left[\mathbb{E}_{n_1,n_2}\left[l_e^\top l_e\right]\right]\mathbb{E}_e\left[\mathbb{E}_{n_1,n_2}\left[2l_e^\top C_1^e\right]\right]
\end{aligned}
\tag{25}
$$

Calculate the derivative by terms:

$$
\begin{aligned}
\mathbb{E}_{n_1,n_2}[l_e^\top l_e] =\ &\mathbb{E}_{n_1,n_2}[C_1^{e\top}C_1^e(\theta_1)^2 + C_1^{e\top}C_2^e\theta_1\theta_2 + C_1^{e\top}Z^e\theta_1\theta_2 - C_1^{e\top}\tilde{A}^{e^k}X_1\theta_1 - C_1^{e\top}n_1\theta_1 \\
&+ C_2^{e\top}C_1^e\theta_1\theta_2 + C_2^{e\top}C_2^e(\theta_2)^2 + C_2^{e\top}Z^e\theta_1\theta_2 - C_2^{e\top}\tilde{A}^{e^k}X_1\theta_2 - C_2^{e\top}n_1\theta_2 \\
&+ Z^{e\top}C_1^e\theta_1\theta_2 + Z^{e\top}C_2^e(\theta_2)^2 + Z^{e\top}Z^e(\theta_2)^2 - Z^{e\top}\tilde{A}^{e^k}X_1\theta_2 - Z^{e\top}n_1\theta_2 \\
&- (\tilde{A}^{e^k}X_1)^\top(C_1^e\theta_1 + C_2^e\theta_2) - (\tilde{A}^{e^k}X_1)^\top Z^e\theta_2 + (\tilde{A}^{e^k}X_1)^\top\tilde{A}^{e^k}X_1 \\
&+ (\tilde{A}^{e^k}X_1)^\top n_1 - n_1^\top(C_1^e\theta_1 + C_2^e\theta_2) - n_1^\top Z^e\theta_2 + n_1^\top\tilde{A}^{e^k}X_1 + n_1^\top n_1]
\end{aligned}
\tag{26}
$$

Since $n_1$ and $n_2$ are independent standard Gaussian noise, we have $\mathbb{E}_{n_1,n_2}[n_1] = \mathbb{E}_{n_1,n_2}[n_2] = \mathbf{0}$, $\mathbb{E}_{n_1,n_2}[n_1^\top n_2] = \mathbb{E}_{n_1,n_2}[n_2^\top n_1] = 0$ and $\mathbb{E}_{n_1,n_2}[n_1^\top n_1] = \mathbb{E}_{n_1,n_2}[n_2^\top n_2] = N^e$ if it is the noise from $e$. Also, since $\epsilon^e$ and $n_1$, $n_2$ are independent, we have $\mathbb{E}_{n_1,n_2}[n_1^\top\epsilon^e] = \mathbb{E}_{n_1,n_2}[n_2^\top\epsilon^e] = 0$.

When

$$
\begin{cases}
\theta_1^{1(l)} = 1, \theta_1^{2(l)} = 1, & l = L-1, ..., L-s+1 \\
\theta_1^{1(l)} = 0, \theta_1^{2(l)} = 1, & l = L-s, L-s-1, ..., 1 \\
\theta_2^{1(l)} = 0, \theta_2^{2(l)} = 1, & l = L-1, ..., 1
\end{cases},
\tag{27}
$$

we have $C_2^{e'} = I_{N^e} \in \mathbb{R}^{N^e \times N^e}$ and $C_1^e = \tilde{A}^{e^s}X_1$. Consequently, we get $\mathbb{E}_{n_1,n_2}[Z^{e\top}n_1] = \text{tr}(C_2^{e'}\tilde{A}^{e^k}) = \text{tr}(\tilde{A}^{e^k})$, $\mathbb{E}_{n_1,n_2}[Z^{e\top}Z^e] = \text{tr}\left((\tilde{A}^{e^k})^\top(\tilde{A}^{e^k})\right) + N^e + \epsilon^{e\top}\epsilon^e$.

Use the above conclusions and rewrite Equation (26) as (here we only plug in the value of $C_2^{e'}$):

$$
\begin{aligned}
&\mathbb{E}_{n_1,n_2}[l_e^\top l_e] = \\
&\left.\begin{array}{l}
C_1^{e\top}C_1^e(\theta_1)^2 + C_1^{e\top}C_2^e\theta_1\theta_2 - C_1^{e\top}\tilde{A}^{e^k}X_1\theta_1 + C_2^{e\top}C_1^e\theta_1\theta_2 + C_2^{e\top}C_2^e(\theta_2)^2 - C_2^{e\top}\tilde{A}^{e^k}X_1\theta_2 \\
+\text{tr}\left((\tilde{A}^{e^k})^\top(\tilde{A}^{e^k})\right)(\theta_2)^2 - (\tilde{A}^{e^k}X_1)^\top(C_1^e\theta_1 + C_2^e\theta_2) + (\tilde{A}^{e^k}X_1)^\top\tilde{A}^{e^k}X_1 + N^e\left(1 + (\theta_2)^2\right) \\
-2\text{tr}(\tilde{A}^{e^k})
\end{array}\right\}(*) \\
&+\left.[C_1^{e\top}\epsilon^e + C_2^{e\top}\epsilon^e + \epsilon^{e\top}C_1^e]\theta_1\theta_2 + \epsilon^{e\top}\epsilon^e(\theta_2)^2 - 2(\tilde{A}^{e^k}X_1)^\top\epsilon^e\theta_2\right\}(**),
\end{aligned}
\tag{28}
$$

$(*)$ and $(**)$ represent terms that are independent and associated with $\epsilon^e$, respectively. Additionally,

$$
\begin{aligned}
\mathbb{E}_{n_1,n_2}[2l_e^\top C_1^e] &= 2\left[C_1^{e\top}C_1^e\theta_1 + C_2^{e\top}C_1^e\theta_2 + (C_2^{e'}\epsilon^e)^\top C_1^e\theta_2 - (\tilde{A}^{e^k}X_1)^\top C_1^e\right] \\
&= 2\left[C_1^{e\top}C_1^e\theta_1 + C_2^{e\top}C_1^e\theta_2 + \epsilon^{e\top}C_1^e\theta_2 - (\tilde{A}^{e^k}X_1)^\top C_1^e\right].
\end{aligned}
\tag{29}
$$

Multiplying Equation (28) and (29) and take the expectation on $e$, using the assumption that $\mathbb{E}_e[(\epsilon^e_i)^2] = \sigma^2$ ($\epsilon^e_i$ is the $i$-th element of $\epsilon^e$):

$$\mathbb{E}_e\left[2\mathbb{E}_{n_1,n_2}[l_e^\top l_e]\mathbb{E}_{n_1,n_2}\left[2l_e^\top C_1\right]\right] = 4\mathbb{E}_e\left[(*)\left(C_1^{e\top}C_1^e\theta_1 + C_2^{e\top}C_2^{e\prime}\theta_2 - (\tilde{A}^{e^k}X_1)^\top C_1^e\right)\right]$$
$$+ 4\mathbb{E}_e\left[(\tilde{A}^{e^s}X_1)^\top \tilde{A}^{e^s}X_1(3\theta_1\theta_2 + (\theta_2)^2) - 2(\tilde{A}^{e^s}X_1)^\top(\tilde{A}^{e^k}X_1)\theta_2\right]\theta_2\sigma^2$$
$$+ 4\mathbb{E}_e[N^e\epsilon^{e\top}\epsilon^e\epsilon^{e\top}(\tilde{A}^{e^s}X_1)]\theta_2.$$

$$(30)$$

Next target is to compute $2\mathbb{E}_e[\mathbb{E}_{n_1,n_2}[l_e^\top l_e]]$ and $\mathbb{E}_e[\mathbb{E}_{n_1,n_2}[2l_e^\top C_1]]$ Since $\epsilon^e$ has zero mean, we have:

$$2\mathbb{E}_e[\mathbb{E}_{n_1,n_2}[l_e^\top l_e]] = \mathbb{E}_e[(*)] + 2\mathbb{E}_e[N^e](\theta_2)^2\sigma^2 \qquad (31)$$

and

$$\mathbb{E}_e[\mathbb{E}_{n_1,n_2}[2l_e^\top C_1^e]] = 2\mathbb{E}_e\left[C_1^{e\top}C_1^e\theta_1 + C_2^{e\top}C_1^e\theta_2 - (\tilde{A}^{e^k}X_1)^\top C_1^e\right]. \qquad (32)$$

Use Equation (30) (31) and (32) and let $\frac{\partial \mathbb{V}_e[R(e)]}{\partial \theta_1} = 0$, we have:

$$\mathbb{E}_e\left[3(\tilde{A}^{e^s}X_1)^\top(\tilde{A}^{e^s}X_1)(\theta_1\theta_2 + \frac{1}{3}(\theta_2)^2) - 2(\tilde{A}^{e^s}X_1)^\top(\tilde{A}^{e^k}X_1)\theta_2\right]\sigma^2 + \mathbb{E}_e[\epsilon^{e\top}\epsilon^e\epsilon^{e\top}(\tilde{A}^{e^s}X_1)]$$
$$-\mathbb{E}_e[N^e]\mathbb{E}_e\left[2(\tilde{A}^{e^s}X_1)^\top(\tilde{A}^{e^s}X_1)(\theta_1 + \theta_2) - (\tilde{A}^{e^k}X_1)^\top C_1^e\right]\theta_2\sigma^2 = 0.$$

$$(33)$$

Now we start calculating the expression of $\frac{\partial \mathbb{V}_e[R(e)]}{\partial \theta_2}$:

$$\frac{\partial \mathbb{V}_e[R(e)]}{\partial \theta_2} = \mathbb{E}_e\left[2\mathbb{E}_{n_1,n_2}\left[l_e^\top l_e\right]\mathbb{E}_{n_1,n_2}\left[2l_e^\top(C_2 + Z^e)\right]\right]$$
$$- 2\mathbb{E}_e\left[\mathbb{E}_{n_1,n_2}\left[l_e^\top l_e\right]\right]\mathbb{E}_e\left[\mathbb{E}_{n_1,n_2}\left[2l_e^\top(C_2^e + Z^e)\right]\right]. \qquad (34)$$

Let $\frac{\partial \mathbb{V}_e[R(e)]}{\partial \theta_2} = 0$:

$$\mathbb{E}_e\left[(C_1^{e\top}C_2^{e\prime} + C_2^{e\top}C_2^{e\prime} + C_2^{e\prime\top}C_1^{e\top})\theta_1\theta_2 + (C_2^{e\prime})^\top C_2^{e\prime}(\theta_2)^2 - 2(\tilde{A}^{e^k}X_1)^\top C_2^{e\prime}\theta_2\right]$$
$$\mathbb{E}_e\left[(C_2^{e\prime\top}C_2^e\theta_2 - (\tilde{A}^{e^k}X_1)^\top C_2^{e\prime})\right]\sigma^2$$
$$- \mathbb{E}_e\left[N^e\sigma^2\left(C_1^{e\top}C_2^e\theta_1 + C_2^{e\top}C_2^e\theta_2 - (\tilde{A}^{e^k}X_1)^\top C_2^e + \mathrm{tr}((\tilde{A}^{e^k})^\top\tilde{A}^{e^k}) + N^e + C_2^{e\prime\top}C_2^{e\prime}\sigma^2\right)(\theta_2)^2\right]$$
$$= 0.$$

$$(35)$$

Plug Equation (33) in (35), we reach:

$$\left[\mathbb{E}_e\left[N^e(\tilde{A}^{e^s}X_1)^\top(\tilde{A}^{e^s}X_1)(\theta_1 + \theta_2) - (\tilde{A}^{e^k}X_1)^\top C_1^e\right]\theta_2\sigma^2 - \mathbb{E}_e[\epsilon^{e\top}\epsilon^e\epsilon^{e\top}(\tilde{A}^{e^s}X_1)]\right]$$
$$\mathbb{E}_e\left((\tilde{A}^{e^s}X_1)^\top\mathbf{1}_{\mathbf{N}^e}\theta_2 - (\tilde{A}^{e^k}X_1)^\top\mathbf{1}_{\mathbf{N}^e}\right)$$
$$- \mathbb{E}_e\left[N^e\left((\tilde{A}^{e^s}X_1)^\top\tilde{A}^{e^s}X_1(\theta_1 + \theta_2) - (\tilde{A}^{e^k}X_1)^\top\tilde{A}^{e^s}X_1 + \mathrm{tr}((\tilde{A}^{e^k})^\top\tilde{A}^{e^k}) + N^e(1 + \sigma^2)\right)\right](\theta_2)^2$$
$$= 0.$$

$$(36)$$

Let $c_1 = \mathbb{E}_e[(\tilde{A}^{e^s}X_1)^\top(\tilde{A}^{e^s}X_1)]$, $c_2 = \mathbb{E}_e[N_e(\tilde{A}^{e^s}X_1)^\top\tilde{A}^{e^s}X_1]$, $c_3 = \mathbb{E}_e[(\tilde{A}^{e^s}X_1)^\top\mathbf{1}]$, $c_4 = \mathbb{E}_e[((\tilde{A}^{e^k}X_1)^\top\mathbf{1}]$, $c_5 = \mathbb{E}_e[N_e((\tilde{A}^{e^k}X_1)^\top\tilde{A}^{e^s}X_1 + \mathrm{tr}((\tilde{A}^{e^k})^\top\tilde{A}^{e^k}) + N^e(1 + \sigma^2))]$, $c_6 = \mathbb{E}_e[(\tilde{A}^{e^s}X_1)^\top(\tilde{A}^{e^k}X_1)]$, $c_7 = \mathbb{E}_e[\epsilon^{e\top}\epsilon^e\epsilon^{e\top}(\tilde{A}^{e^s}X_1)]$,

we conclude that

$$
\begin{cases}
(3c_1\theta_1\theta_2 + c_1(\theta_2)^2 - 2c_6\theta_2)\sigma^2 - \mathbb{E}_e[N^e(2c_1(\theta_1+\theta_2) - c_6)]\sigma^2\theta_2 + c_7 = 0 \\
(\mathbb{E}_e[N^e(2c_1(\theta_1+\theta_2) - c_6)]\sigma^2\theta_2 - c_7)(c_3\theta_2 - c_4) - [c2(\theta_1+\theta_2) - c_5](\theta_2)^2 = 0
\end{cases} \cdot \quad (37)
$$

As for the derivative respect to $\theta_1^{1\,(l)}$, $\theta_1^{2\,(l)}$, $\theta_2^{1\,(l)}$, $\theta_2^{2\,(l)}$, when they take the special value in (27), we have $\frac{\partial \mathbb{V}_e[R(e)]}{\partial \theta_1} \Rightarrow \theta_1^{1\,(l)} = \theta_1^{2\,(l)} = 0$ and $\frac{\partial \mathbb{V}_e[R(e)]}{\partial \theta_2} \Rightarrow \theta_2^{1\,(l)} = \theta_2^{2\,(l)} = 0, l = 1, ..., L$ So we conclude the solution induced by 37 is the solution of the objective.

$\square$

### E.1.2 Proof of the Failure Case of IRMv1 under Concept Shift

**Proposition E.2.** *(IRMv1 will use spurious features)* *The objective* $\min_\Theta \mathbb{E}_e[\|\nabla_{w|w=1.0}R(e)\|^2]$ *has a solution that uses spurious features:*

$$
\begin{cases}
\theta_1 = \frac{\mathbb{E}_e\left\{(\tilde{A}^{e^s}X_1)^\top(\tilde{A}^{e^k}\mathbf{1})\left[\mathbf{1}^\top\tilde{A}^{e^s}X_1 + (\tilde{A}^{e^k}X_1)^\top(\tilde{A}^{e^k}\mathbf{1})\right] + (1+\sigma^2)(\tilde{A}^{e^s}X_1)^\top\mathbf{1}(\tilde{A}^{e^k}X_1)^\top\mathbf{1}\right\}}{(2+\sigma^2)(\mathbb{E}_e[\tilde{A}^{e^s}]X_1)^\top\mathbf{1}} \\
\theta_2 = \frac{\mathbb{E}_e\left\{(\tilde{A}^{e^s}X_1)^\top(\tilde{A}^{e^s}X_1)[\mathbf{1}^\top(\tilde{A}^{e^k}\mathbf{1})] + (\tilde{A}^{e^s}X_1)^\top(\tilde{A}^{e^k}\mathbf{1})(\mathbf{1}^\top\tilde{A}^{e^s}X_1)\right\}}{(2+\sigma^2)(\mathbb{E}_e[\tilde{A}^{e^s}]X_1)^\top\mathbf{1}}
\end{cases} \cdot \quad (38)
$$

*when* $\{\theta_1^{1\,(l)}, \theta_1^{2\,(l)}, \theta_2^{1\,(l)}, \theta_2^{2\,(l)}\}$ *take the special values for some* $0 < s < L$:

$$
\Theta_0 = \begin{cases}
\theta_1^{1\,(l)} = 1, \theta_1^{2\,(l)} = 1, & l = L-1, ..., L-s+1 \\
\theta_1^{1\,(l)} = 0, \theta_1^{2\,(l)} = 1, & l = L-s, L-s-1, ..., 1 \\
\theta_2^{1\,(l)} = 0, \theta_2^{2\,(l)} = 1, & l = L-1, ..., 1
\end{cases} \cdot \quad (39)
$$

*Proof.*

$$
\begin{aligned}
\mathcal{L}_{\text{IRMv1}} &= \mathbb{E}_e\left[\|\nabla_{w|w=1.0}\mathbb{E}_{n_1,n_2}[\mathcal{L}(f_\Theta(A^e, X^e), Y^e)]\|_2^2\right] \\
&= \mathbb{E}_e\left[2\mathbb{E}_{n_1,n_2}[(\hat{Y}^e - Y^e)^\top\phi(A^e, X^e)]\right] \\
&= \mathbb{E}_e\left[2\mathbb{E}_{n_1,n_2}\left[\left(C_1^e\theta_1 + (C_2^e + Z^e)\theta_2 - \tilde{A}^{e^k}X_1 - n_1\right)^\top(C_1^e\theta_1 + (C_2^e + Z^e)\theta_2)\right]\right] \\
&= \mathbb{E}_e\left[2\mathbb{E}_{n_1,n_2}\left[C_1^{e\top}C_1^e(\theta_1)^2 + 2C_1^{e\top}(C_2^e + Z^e)\theta_1\theta_2 + C_2^{e\top}C_2^e(\theta_2)^2\right]\right] \\
&\quad - \mathbb{E}_e\left[2\mathbb{E}_{n_1,n_2}\left[(\tilde{A}^{e^k}X_1 + n_1)^\top C_1^e\theta_1 - ((\tilde{A}^{e^k}X_1) + n_1)^\top(C_2^e + Z^e)\theta_2\right]\right].
\end{aligned} \quad (40)
$$

Take the derivative of $\mathcal{L}_{\text{IRMv1}}$ w.r.t. $\theta_1$ and $\theta_2$:

$$
\begin{cases}
\frac{\partial \mathcal{L}_{\text{IRMv1}}}{\partial \theta_1} = \mathbb{E}_e[2\mathbb{E}_{n_1,n_2}[C_1^{e\top}C_1^e\theta_1 + 2C_1^{e\top}(C_2^e + Z^e)\theta_2 - ((\tilde{A}^{e^k}X_1) + n_1)^\top C_1^e]] \\
\frac{\partial \mathcal{L}_{\text{IRMv1}}}{\partial \theta_2} = \mathbb{E}_e[2\mathbb{E}_{n_1,n_2}[C_2^{e\top}C_2^e\theta_2 + 2C_1^{e\top}(C_2^e + Z^e)\theta_1 - ((\tilde{A}^{e^k}X_1) + n_1)^\top(C_2^e + Z^e)^e]]
\end{cases} \cdot \quad (41)
$$

For brevity, let $a = C_1^{e\top}C_1^e\theta_1$, $b = C_1^{e\top}(C_2^e + Z^e)$, $c = ((\tilde{A}^{e^k}X_1) + n_1)^\top C_1^e$, $d = C_2^{e\top}C_2^e$, $e = ((\tilde{A}^{e^k}X_1) + n_1)^\top(C_2^e + Z^e)^e$. By letting $\{\theta_1^{1\,(l)}, \theta_1^{2\,(l)}, \theta_2^{1\,(l)}, \theta_2^{2\,(l)}\}$ take the values of $\Theta_0$, let the derivative w.r.t. $\theta_1$ and $\theta_2$ to be zero, we have

$$
\begin{cases}
\frac{\partial \mathcal{L}_{\text{IRMv1}}}{\partial \theta_1} = \frac{ac - be}{2(a^2 - b^2)} \\
\frac{\partial \mathcal{L}_{\text{IRMv1}}}{\partial \theta_2} = \frac{ae - bc}{2(a^2 - b^2)}
\end{cases} \cdot \quad (42)
$$

Also, when $\{\theta_1^{1\,(l)}, \theta_1^{2\,(l)}, \theta_2^{1\,(l)}, \theta_2^{2\,(l)}\}$ take the values of $\Theta_0$, we have

$$
\frac{\partial \mathcal{L}_{\text{IRMv1}}}{\partial \theta_1} = \frac{\partial \mathcal{L}_{\text{IRMv1}}}{\partial \theta_1^{1\,(l)}} = \frac{\partial \mathcal{L}_{\text{IRMv1}}}{\partial \theta_1^{2\,(l)}} = 0
$$

$$
\frac{\partial \mathcal{L}_{\text{IRMv1}}}{\partial \theta_2} = \frac{\partial \mathcal{L}_{\text{IRMv1}}}{\partial \theta_2^{1\,(l)}} = \frac{\partial \mathcal{L}_{\text{IRMv1}}}{\partial \theta_2^{2\,(l)}} = 0
$$

$$(43)$$

$\square$

### E.1.3 PROOF OF THE SUCCESSFUL CASE OF CIA UNDER CONCEPT SHIFT

**Proposition E.3.** *Optimizing the CIA objective will lead to the optimal solution $\Theta^*$:*

$$
\begin{cases}
\theta_1 = 1 \\
\theta_2 = 0 \quad or \quad \exists l \in \{1, ..., L-1\} \ s.t. \ {\theta_2^1}^{(l)} = {\theta_2^2}^{(l)} = 0 \\
{\theta_1^1}^{(l)} = 1, {\theta_1^2}^{(l)} = 1, \quad l = L-1, ..., L-k+1 \\
{\theta_1^1}^{(l)} = 0, {\theta_1^2}^{(l)} = 1, \quad l = L-k, L-k-1, ..., 1
\end{cases} \tag{44}
$$

*Proof.* For brevity, denote a node representation of $C_{1\,c}^e$ as $C_1^i$ and the one of $C_{1\,c}^{e'}$ as $C_1^j$. The same is true for $C_2^i$ and $C_2^j$. In this toy model, we need to consider the expectation of the noise, while in real cases such noise is included in the node features so taking expectation on $e$ will handle this. Therefore, we add $\mathbb{E}_{n_1,n_2}$ in this proof, and this expectation is excluded in the formal description of the objective in the main paper.

$$
\begin{aligned}
\mathcal{L}_{\mathrm{CIA}} &= \mathbb{E}_{\substack{e,e' \\ e \neq e'}} \mathbb{E}_{n_1,n_2} \mathbb{E}_c \mathbb{E}_{\substack{i,j \\ (i,j) \in \Omega^{e,e'}}} \left[ \mathcal{D}(\phi_\Theta(A^e, X^e)_{[c][v_i]}, \phi_\Theta(A^e, X^{e'})_{[c][v_j]}) \right] \\
&= \mathbb{E}_{\substack{e,e' \\ e \neq e'}} \mathbb{E}_{n_1,n_2} \mathbb{E}_c \mathbb{E}_{\substack{i,j \\ (i,j) \in \Omega^{e,e'}}} \| C_1^i \theta_1 + (C_2^i + Z^e)\theta_2 - C_1^j \theta_1 - (C_2^j + Z^{e'})\theta_2 \|_2^2
\end{aligned} \tag{45}
$$

$$
\frac{\partial \mathcal{L}_{\mathrm{CIA}}}{\partial \theta_1} = \mathbb{E}_{\substack{e,e' \\ e \neq e'}} \mathbb{E}_{n_1,n_2} \mathbb{E}_c \mathbb{E}_{\substack{i,j \\ (i,j) \in \Omega^{e,e'}}} \left[ C_1^i \theta_1 + (C_2^i + Z^e)\theta_2 - C_1^j \theta_1 - (C_2^j + Z^{e'})\theta_2 \right]^\top (C_1^i - C_1^j) \tag{46}
$$

Let $\frac{\partial \mathcal{L}_{\mathrm{CIA}}}{\partial \theta_1} = 0$, we have:

$$
\mathbb{E}_{\substack{e,e' \\ e \neq e'}} \mathbb{E}_c \mathbb{E}_{\substack{i,j \\ (i,j) \in \Omega^{e,e'}}} \left[ (C_1^i - C_1^j)^\top (C_1^i - C_1^j)\theta_1 + (C_2^i - C_2^k)^\top (C_1^i - C_1^j)\theta_2 \right] = 0 \tag{47}
$$

Also, we have:

$$
\frac{\partial \mathcal{L}_{\mathrm{CIA}}}{\partial \theta_2} = \mathbb{E}_{\substack{e,e' \\ e \neq e'}} \mathbb{E}_c \mathbb{E}_{\substack{i,j \\ (i,j) \in \Omega^{e,e'}}} \left[ (C_1^i - C_1^j)^\top (C_2^i - C_2^j)\theta_1 + \left[ (C_2^i - C_2^k)^\top (C_2^i - C_2^j) + (Z^e - Z^{e'})^\top (Z^e - Z^{e'}) \right] \theta_2 \right] \tag{48}
$$

Further let $\frac{\partial \mathcal{L}_{\mathrm{CIA}}}{\partial \theta_2} = 0$, combining Equation (47), we get

$$
\begin{cases}
\theta_1 = 0 \quad or \quad \exists l \in \{1, ..., L-1\} \ s.t. \ {\theta_1^1}^{(l)} = {\theta_1^2}^{(l)} = 0 \\
\theta_2 = 0
\end{cases} \tag{49}
$$

or, if $\theta_1 \neq 0$ and $\forall l \in \{1, ..., L-1\}$, the parameters of that layer $l$ of the invariant branch of the GNN are not all zero: ${\theta_1^1}^{(l)} \neq 0$ or ${\theta_1^2}^{(l)} \neq 0$, then we get

$$
\theta_2 \underbrace{\mathbb{E}_{\substack{e,e' \\ e \neq e'}} \mathbb{E}_c \mathbb{E}_{\substack{i,j \\ (i,j) \in \Omega^{e,e'}}} \left[ -\frac{[(C_1^i - C_1^j)^\top (C_2^i - C_2^j)]^2}{(C_1^i - C_1^j)^\top (C_1^i - C_1^j)} + (C_2^i - C_2^j)^\top (C_2^i - C_2^j) + (Z^e - Z^{e'})^\top (Z^e - Z^{e'}) \right]}_{F} = 0 \tag{50}
$$

Due to the property of the inner product, $F > 0$ unless $\exists l \in \{1, ..., L-1\}$ s.t. ${\theta_2^1}^{(l)} = {\theta_2^2}^{(l)} = 0$. To ensure $\frac{\partial \mathcal{L}_{\mathrm{CIA}}}{\partial \theta_2}$, we conclude that $\theta_2 = 0$ or $\exists l \in \{1, ..., L-1\}$ s.t. ${\theta_2^1}^{(l)} = {\theta_2^2}^{(l)} = 0$.

In conclusion, to satisfy the constraint of CIA, no matter whether the invariant branch has zero output, the spurious branch must have zero parameters, i.e.,

$$
\theta_2 = 0 \quad or \quad \exists l \in \{1, ..., L-1\} \ s.t. \ {\theta_2^1}^{(l)} = {\theta_2^2}^{(l)} = 0 \tag{51}
$$

Thus, CIA will remove spurious features.

Now we show that when CIA objective has been reached (the spurious branch has zero outputs), the objective of $\min_\Theta \mathbb{E}_e[\mathcal{L}(f_\Theta(A^e, X^e), Y^e)]$ will help to learn predictive paramters of the invariant branch $\theta_1$, $\theta_1^{1(l)}$ and $\theta_1^{2(l)}$. When Equation (51) holds,

$$
\begin{aligned}
\frac{\partial \mathbb{E}_e[\mathcal{L}(f_\Theta(A^e, X^e), Y^e)]}{\partial \theta_1} &= 2\mathbb{E}_e \mathbb{E}_{n_1, n_2}\left[ \left( C_1^e \theta_1 - \tilde{A}^{e^k} X_1 - n_1 \right)^\top C_1^e \right] \\
&= 2\mathbb{E}_e \left[ \left( C_1^e \theta_1 - \tilde{A}^{e^k} X_1 \right)^\top C_1^e \right]
\end{aligned}
\tag{52}
$$

Let $\frac{\partial \mathbb{E}_e[\mathcal{L}(f_\Theta(A^e, X^e), Y^e)]}{\partial \theta_1} = 0$, we get the predictive parameters

$$
\begin{cases}
\theta_1 = 1 \\
\theta_1^{1(l)} = 1, \theta_1^{2(l)} = 1, & l = L-1, ..., L-k+1 \\
\theta_1^{1(l)} = 0, \theta_1^{2(l)} = 1, & l = L-k, L-k-1, ..., 1
\end{cases}
\tag{53}
$$

Plug the final solution back in $\frac{\partial \mathcal{L}_{\text{CIA}}}{\partial \theta_1^{1(l)}}$, $\frac{\partial \mathcal{L}_{\text{CIA}}}{\partial \theta_1^{2(l)}}$, $\frac{\partial \mathcal{L}_{\text{CIA}}}{\partial \theta_2^{1(l)}}$, $\frac{\partial \mathcal{L}_{\text{CIA}}}{\partial \theta_2^{2(l)}}$, $\frac{\partial \mathbb{E}_e[\mathcal{L}(f_\Theta(A^e, X^e), Y^e)]}{\partial \theta_1^{1(l)}}$, $\frac{\partial \mathbb{E}_e[\mathcal{L}(f_\Theta(A^e, X^e), Y^e)]}{\partial \theta_1^{2(l)}}$, $\frac{\partial \mathbb{E}_e[\mathcal{L}(f_\Theta(A^e, X^e), Y^e)]}{\partial \theta_2^{1(l)}}$, $\frac{\partial \mathbb{E}_e[\mathcal{L}(f_\Theta(A^e, X^e), Y^e)]}{\partial \theta_2^{2(l)}}$, we can verify that these terms are all 0. □

## E.2 Proof of the Covariate Shift Case

### E.2.1 Proof of the Failure Case of VREx under Covariate Shift

*Proof.* We will use some symbols to simplify the expression of the toy GNN. Denote $n_2 + \epsilon^e$ as $\eta$. Use the following notations to represent the components of the $L$-layer GNN model:

$$
\begin{aligned}
f_\Theta(A, X) &= H_1^{(L)} \theta_1 + H_2^{(L)} \theta_2 \\
&= \underbrace{\left[ \theta_1^{1(L-1)} \bar{A} \left( ...\theta_1^{1(3)} \left( \theta_1^{1(2)} \bar{A}(\theta_1^{1(1)} \bar{A} + \theta_1^{2(1)} \bar{I}) X_1 + \theta_1^{2(2)} (\theta_1^{1(1)} \bar{A} + \theta_1^{2(1)} \bar{I}) X_1 \right) + ... \right) \right]}_{C_1} \theta_1 \\
&+ \underbrace{\left[ \theta_2^{1(L-1)} \bar{A} \left( ...\theta_2^{1(3)} \left( \theta_2^{1(2)} \bar{A}(\theta_2^{1(1)} \bar{A} + \theta_2^{2(1)} \bar{I}) \eta + \theta_2^{2(2)} (\theta_2^{1(1)} \bar{A} + \theta_2^{2(1)} \bar{I}) \eta \right) + ... \right) \right]}_{Z} \theta_2 \\
&= C_1 \theta_1 + Z \theta_2.
\end{aligned}
\tag{54}
$$

$C_1, Z \in \mathbb{R}^{N \times 1}$. We use $C_1^e$ and $Z^e$ to denote the variables from the corresponding environment $e$. We further denote $Z^e = C_2^{e'} \eta$.

Using these notations, the loss of environment $e$ is

$$
\begin{aligned}
R(e) &= \mathbb{E}_{n_1, n_2}\left[ \|f_\Theta(A^e, X^e) - Y^e\|_2^2 \right] \\
&= \mathbb{E}_{n_1, n_2}\left[ \left\| C_1^e \theta_1 + Z^e \theta_2 - \tilde{A}^{e^k} X_1 - n_1 \right\|_2^2 \right].
\end{aligned}
\tag{55}
$$

Denote the inner term $C_1^e \theta_1 + Z^e \theta_2 - \tilde{A}^{e^k} X_1 - n_1$ as $l_e$.

The variance of loss across environments is:

$$
\begin{aligned}
\mathbb{V}_e[R(e)] &= \mathbb{E}_e[R^2(e)] - \mathbb{E}_e^2[R(e)] \\
&= \mathbb{E}_e\left[ \left( \mathbb{E}_{n_1, n_2} \left\| C_1^e \theta_1 + Z^e \theta_2 - \tilde{A}^{e^k} X_1 - n_1 \right\|_2^2 \right)^2 \right] \\
&\quad - \mathbb{E}_e^2 \left[ \mathbb{E}_{n_1, n_2} \left\| C_1^e \theta_1 + Z^e \theta_2 - \tilde{A}^{e^k} X_1 - n_1 \right\|_2^2 \right] . \\
&= \mathbb{E}_e\left[ \mathbb{E}_{n_1, n_2}\left[ (l_e^\top l_e)^2 \right] \right] - \mathbb{E}_e^2 \left[ \mathbb{E}_{n_1, n_2}\left[ l_e^\top l_e \right] \right].
\end{aligned}
\tag{56}
$$

Take the derivative of $\mathbb{V}_e[R(e)]$ with respect to $\theta_1$:

$$
\begin{aligned}
\frac{\partial \mathbb{V}_e[R(e)]}{\partial \theta_1} = {}& \mathbb{E}_e \left[ 2\mathbb{E}_{n_1,n_2} \left[ l_e^\top l_e \right] \mathbb{E}_{n_1,n_2} \left[ 2l_e^\top C_1^e \right] \right] \\
& - 2\mathbb{E}_e \left[ \mathbb{E}_{n_1,n_2} \left[ l_e^\top l_e \right] \right] \mathbb{E}_e \left[ \mathbb{E}_{n_1,n_2} \left[ 2l_e^\top C_1^e \right] \right]
\end{aligned}
\tag{57}
$$

Calculate the derivative by terms:

$$
\begin{aligned}
\mathbb{E}_{n_1,n_2}[l_e^\top l_e] = \mathbb{E}_{n_1,n_2} \big[ & C_1^{e\top} C_1^e (\theta_1)^2 + C_1^{e\top} Z^e \theta_1 \theta_2 - C_1^{e\top} \tilde{A}e^k X_1 \theta_1 - C_1^{e\top} n_1 \theta_1 \\
& + Z^{e\top} C_1^e \theta_1 \theta_2 + Z^{e\top} Z^e (\theta_2)^2 - Z^{e\top} \tilde{A}e^k X_1 \theta_2 - Z^{e\top} n_1 \theta_2 \\
& - (\tilde{A}e^k X_1)^\top C_1^e \theta_1 - (\tilde{A}e^k X_1)^\top Z^e \theta_2 + (\tilde{A}e^k X_1)^\top \tilde{A}e^k X_1 \\
& + (\tilde{A}e^k X_1)^\top n_1 - n_1^\top C_1^e \theta_1 - n_1^\top Z^e \theta_2 + n_1^\top \tilde{A}e^k X_1 + n_1^\top n_1 \big]
\end{aligned}
\tag{58}
$$

Since $n_1$ and $n_2$ are independent standard Gaussian noise, we have $\mathbb{E}_{n_1,n_2}[n_1] = \mathbb{E}_{n_1,n_2}[n_2] = \mathbf{0}$, $\mathbb{E}_{n_1,n_2}[n_1^\top n_2] = \mathbb{E}_{n_1,n_2}[n_2^\top n_1] = 0$ and $\mathbb{E}_{n_1,n_2}[n_1^\top n_1] = \mathbb{E}_{n_1,n_2}[n_2^\top n_2] = N^e$ if it is the noise from $e$. Also, since $\epsilon^e$ and $n_1$, $n_2$ are independent, we have $\mathbb{E}_{n_1,n_2}[n_1^\top \epsilon^e] = \mathbb{E}_{n_1,n_2}[n_2^\top \epsilon^e] = 0$.

When

$$
\begin{cases}
\theta_1^{1\,(l)} = 1, \theta_1^{2\,(l)} = 1, & l = L-1, ..., L-s+1 \\
\theta_1^{1\,(l)} = 0, \theta_1^{2\,(l)} = 1, & l = L-s, L-s-1, ..., 1 \\
\theta_2^{1\,(l)} = 0, \theta_2^{2\,(l)} = 1, & l = L-1, ..., 1
\end{cases}
,
\tag{59}
$$

we have $C_2^{e\prime} = I_{N^e} \in \mathbb{R}^{N^e \times N^e}$ and $C_1^e = \tilde{A}e^s X_1$. Consequently, we get $\mathbb{E}_{n_1,n_2}[Z^{e\top} n_1] = 0$, $\mathbb{E}_{n_1,n_2}[Z^{e\top} Z^e] = N^e + \epsilon^{e\top} \epsilon^e$.

Use the above conclusions and rewrite Equation (58) as (here we only plug in the value of $C_2^{e\prime}$):

$$
\begin{aligned}
\mathbb{E}_{n_1,n_2}[l_e^\top l_e] = {}& \\
& \left. \begin{aligned} & C_1^{e\top} C_1^e (\theta_1)^2 - C_1^{e\top} \tilde{A}e^k X_1 \theta_1 \\ & - (\tilde{A}e^k X_1)^\top C_1^e \theta_1 + (\tilde{A}e^k X_1)^\top \tilde{A}e^k X_1 + N^e \left( 1 + (\theta_2)^2 \right) \end{aligned} \right\} (*) \\
& + \left. \left[ C_1^{e\top} \epsilon^e + \epsilon^{e\top} C_1^e \right] \theta_1 \theta_2 + \epsilon^{e\top} \epsilon^e (\theta_2)^2 - 2(\tilde{A}e^k X_1)^\top \epsilon^e \theta_2 \right\} (**),
\end{aligned}
\tag{60}
$$

$(*)$ and $(**)$ represent terms that are independent and associated with $\epsilon^e$, respectively.

Additionally,

$$
\mathbb{E}_{n_1,n_2}[2l_e^\top C_1^e] = 2 \left[ C_1^{e\top} C_1^e \theta_1 + \epsilon^{e\top} C_1^e \theta_2 - (\tilde{A}e^k X_1)^\top C_1^e \right]
\tag{61}
$$

.

Multiplying Equation (60) and (61) and take the expectation on $e$, using the assumption that $\mathbb{E}_e[(\epsilon^e{}_i)^2] = \sigma^2$ ($\epsilon^e{}_i$ is the $i$-th element of $\epsilon^e$):

$$
\begin{aligned}
\mathbb{E}_e \left[ 2\mathbb{E}_{n_1,n_2}[l_e^\top l_e] \mathbb{E}_{n_1,n_2} \left[ 2l_e^\top C_1 \right] \right] = {}& 4\mathbb{E}_e \left[ (*) \left( C_1^{e\top} C_1^e \theta_1 - (\tilde{A}e^k X_1)^\top C_1^e \right) \right] \\
& + 4\mathbb{E}_e \left[ (\tilde{A}e^s X_1)^\top \tilde{A}e^s X_1 (2\theta_1\theta_2 + (\theta_2)^2) - 2(\tilde{A}e^s X_1)^\top (\tilde{A}e^k X_1)\theta_2 \right] \theta_2 \sigma^2 \\
& + 4\mathbb{E}_e[N^e \epsilon^{e\top} \epsilon^e \epsilon^{e\top} (\tilde{A}e^s X_1)]\theta_2.
\end{aligned}
\tag{62}
$$

Next target is to compute $2\mathbb{E}_e[\mathbb{E}_{n_1,n_2}[l_e^\top l_e]]$ and $\mathbb{E}_e[\mathbb{E}_{n_1,n_2}[2l_e^\top C_1]]$ Since $\epsilon^e$ has zero mean, we have:

$$
2\mathbb{E}_e[\mathbb{E}_{n_1,n_2}[l_e^\top l_e]] = \mathbb{E}[(*)] + \mathbb{E}[2N^e]\sigma^2 (\theta_2)^2
\tag{63}
$$

and

$$
\mathbb{E}_e[\mathbb{E}_{n_1,n_2}[2l_e^\top C_1^e]] = 2\mathbb{E}_e \left[ C_1^{e\top} C_1^e \theta_1 - (\tilde{A}e^k X_1)^\top C_1^e \right].
\tag{64}
$$

Use Equation (62) (63) and (64) and let $\frac{\partial \mathbb{V}_e[R(e)]}{\partial \theta_1} = 0$, we have:

$$\mathbb{E}_e \left[ 2(\tilde{A}^{e^s} X_1)^\top (\tilde{A}^{e^s} X_1)(\theta_1 \theta_2 + \frac{1}{2}(\theta_2)^2) - 2(\tilde{A}^{e^s} X_1)^\top (\tilde{A}^{e^k} X_1)\theta_2 \right] \sigma^2 + \mathbb{E}_e[\epsilon^{e^\top} \epsilon^e \epsilon^{e^\top} (\tilde{A}^{e^s} X_1)]$$

$$-\mathbb{E}_e[N^e]\mathbb{E}_e \left[ (\tilde{A}^{e^s} X_1)^\top (\tilde{A}^{e^s} X_1)\theta_1 - (\tilde{A}^{e^k} X_1)^\top C_1^e \right] \theta_2 \sigma^2 = 0. \tag{65}$$

Now we start calculating the expression of $\frac{\partial \mathbb{V}_e[R(e)]}{\partial \theta_2}$:

$$\frac{\partial \mathbb{V}_e[R(e)]}{\partial \theta_2} = \mathbb{E}_e \left[ 2\mathbb{E}_{n_1,n_2} \left[ l_e^\top l_e \right] \mathbb{E}_{n_1,n_2} \left[ 2l_e^\top Z^e \right] \right] \tag{66}$$
$$- 2\mathbb{E}_e \left[ \mathbb{E}_{n_1,n_2} \left[ l_e^\top l_e \right] \right] \mathbb{E}_e \left[ \mathbb{E}_{n_1,n_2} \left[ 2l_e^\top Z^e \right] \right].$$

Let $\frac{\partial \mathbb{V}_e[R(e)]}{\partial \theta_2} = 0$:

$$\mathbb{E}_e \left[ (C_1^{e^\top} C_2^{e'} + C_2^{e'^\top} C_1^{e^\top})\theta_1 \theta_2 + (C_2^{e'})^\top C_2^{e'}(\theta_2)^2 - 2(\tilde{A}^{e^k} X_1)^\top C_2^{e'} \theta_2 \right] \mathbb{E}_e \left[ (-(\tilde{A}^{e^k} X_1)^\top C_2^{e'}) \right] \sigma^2$$

$$- \mathbb{E}_e \left[ N^e \sigma^2 \left( \text{tr}((\tilde{A}^{e^k})^\top \tilde{A}^{e^k}) + N^e + C_2^{e'^\top} C_2^{e'} \sigma^2 \right) (\theta_2)^2 \right]$$

$$= 0. \tag{67}$$

Plug Equation (65) in (67), we reach:

$$\left[ \mathbb{E}_e \left[ N^e (\tilde{A}^{e^s} X_1)^\top (\tilde{A}^{e^s} X_1)\theta_1 - N^e (\tilde{A}^{e^k} X_1)^\top C_1^e \right] \theta_2 \sigma^2 - \mathbb{E}_e[\epsilon^{e^\top} \epsilon^e \epsilon^{e^\top} (\tilde{A}^{e^s} X_1)] \right] \mathbb{E}_e \left( -(\tilde{A}^{e^k} X_1)^\top \mathbf{1_{N^e}} \right)$$

$$- \mathbb{E}_e \left[ N^e \left( \text{tr}((\tilde{A}^{e^k})^\top \tilde{A}^{e^k}) + N^e(1 + \sigma^2) \right) \right] (\theta_2)^2$$

$$= 0. \tag{68}$$

Let $c_1 = \mathbb{E}[(\tilde{A}^{e^s} X_1)^\top (\tilde{A}^{e^s} X_1)]$, $c_2 = \mathbb{E}[(\tilde{A}^{e^s} X_1)^\top (\tilde{A}^{e^k} X_1)]$, $c_3 = \mathbb{E}_e[\epsilon^{e^\top} \epsilon^e \epsilon^{e^\top} (\tilde{A}^{e^s} X_1)]\sigma^2$, $c_4 = \mathbb{E}_e \left[ (\tilde{A}^{e^k} X_1)^\top \mathbf{1_{N^e}} \right] \sigma^2$, $c_5 = \mathbb{E}_e \left[ N^e \left( \text{tr}((\tilde{A}^{e^k})^\top \tilde{A}^{e^k}) + N^e(1 + \sigma^2) \right) \right]$,

we conclude that

$$\begin{cases} c_1 \sigma^2 (2\theta_1 \theta_2 + (\theta_2)^2 - 2c_2 \sigma^2 \theta_2) + c_3 - \mathbb{E}_e[N^e]c_1 \sigma^2 \theta_1 \theta_2 + \mathbb{E}_e[N^e]c_2 \sigma^2 \theta_2 = 0 \\ \left[ c_3 - \mathbb{E}_e[N^e]c_1 \sigma^2 \theta_1 \theta_2 + \mathbb{E}_e[N^e]c_2 \sigma^2 \theta_2 \right] c_4 - c_5 (\theta_2)^2 \end{cases}. \tag{69}$$

As for the derivative respect to $\theta_1^{1^{(l)}}$, $\theta_1^{2^{(l)}}$, $\theta_2^{1^{(l)}}$, $\theta_2^{2^{(l)}}$, when they take the special value in (59), we have $\frac{\partial \mathbb{V}_e[R(e)]}{\partial \theta_1} \Rightarrow \theta_1^{1^{(l)}} = \theta_1^{2^{(l)}} = 0$ and $\frac{\partial \mathbb{V}_e[R(e)]}{\partial \theta_2} \Rightarrow \theta_2^{1^{(l)}} = \theta_2^{2^{(l)}} = 0, l = 1, ..., L$ So we conclude the solution induced by 69 is the solution of the objective. $\square$

### E.2.2 Proof of the Failure Case of IRMv1 under Covariate Shift

*Proof.*

$$\mathcal{L}_{\text{IRMv1}} = \mathbb{E}_e \left[ \| \nabla_{w|w=1.0} \mathbb{E}_{n_1,n_2} [\mathcal{L}(f_\Theta(A^e, X^e), Y^e)] \|_2^2 \right]$$

$$= \mathbb{E}_e \left[ 2\mathbb{E}_{n_1,n_2} [(\hat{Y}^e - Y^e)^\top \phi(A^e, X^e)] \right]$$

$$= \mathbb{E}_e \left[ 2\mathbb{E}_{n_1,n_2} \left[ \left( C_1^e \theta_1 + Z^e \theta_2 - \tilde{A}^{e^k} X_1 - n_1 \right)^\top (C_1^e \theta_1 + Z^e \theta_2) \right] \right] \tag{70}$$

$$= \mathbb{E}_e \left[ 2\mathbb{E}_{n_1,n_2} \left[ C_1^{e^\top} C_1^e (\theta_1)^2 + 2C_1^{e^\top} Z^e \theta_1 \theta_2 \right] \right]$$

$$- \mathbb{E}_e \left[ 2\mathbb{E}_{n_1,n_2} \left[ (\tilde{A}^{e^k} X_1 + n_1)^\top C_1^e \theta_1 - (\tilde{A}^{e^k} X_1 + n_1)^\top Z^e \theta_2 \right] \right].$$

Take the derivative of $\mathcal{L}_{\text{IRMv1}}$ w.r.t. $\theta_1$:

$$\frac{\partial \mathcal{L}_{\text{IRMv1}}}{\partial \theta_1} = \mathbb{E}_e[2\mathbb{E}_{n_1,n_2}[C_1^{e\top}C_1^e\theta_1 + 2C_1^{e\top}Z^e\theta_2 - ((\tilde{A}^{e^k}X_1) + n_1)^\top C_1^e]], \tag{71}$$

Let it be zero, we get $\theta_1 = \frac{\mathbb{E}_e[(\tilde{A}^{e^k}X_1)^\top(\tilde{A}^{e^{2k}}X_1)]}{\mathbb{E}_e[(\tilde{A}^{e^{2k}}X_1)^\top(\tilde{A}^{e^{2k}}X_1)]}$

Take the derivative of $\mathcal{L}_{\text{IRMv1}}$ w.r.t. $\theta_2$:

$$\frac{\partial \mathcal{L}_{\text{IRMv1}}}{\partial \theta_2} = \mathbb{E}_e[2\mathbb{E}_{n_1,n_2}[2C_1^{e\top}Z^e\theta_1 - ((\tilde{A}^{e^k}X_1) + n_1)^\top Z^e]] \equiv 0 \tag{72}$$

Also, when $\{\theta_1^{1(l)}, \theta_1^{2(l)}, \theta_2^{1(l)}, \theta_2^{2(l)}\}$ take the values of $\Theta_0$, we have

$$\frac{\partial \mathcal{L}_{\text{IRMv1}}}{\partial \theta_1} = \frac{\partial \mathcal{L}_{\text{IRMv1}}}{\partial \theta_1^{1(l)}} = \frac{\partial \mathcal{L}_{\text{IRMv1}}}{\partial \theta_1^{2(l)}} = 0$$
$$\frac{\partial \mathcal{L}_{\text{IRMv1}}}{\partial \theta_2} = \frac{\partial \mathcal{L}_{\text{IRMv1}}}{\partial \theta_2^{1(l)}} = \frac{\partial \mathcal{L}_{\text{IRMv1}}}{\partial \theta_2^{2(l)}} = 0 \tag{73}$$

$\square$

### E.3 Proof of the Successful Case of CIA under Covariate Shift

*Proof.* For brevity, denote a node representation of $C_{1_c}^e$ as $C_1^i$ and the one of $C_{1_c}^{e'}$ as $C_1^j$. The same is true for $C_2^i$ and $C_2^j$. In this toy model, we need to consider the expectation of the noise, while in real cases such noise is included in the node features so taking expectation on $e$ will handle this. Therefore, we add $\mathbb{E}_{n_1,n_2}$ in this proof, and this expectation is excluded in the formal description of the objective in the main paper.

$$\mathcal{L}_{\text{CIA}} = \mathbb{E}_{\substack{e,e' \\ e \neq e'}} \mathbb{E}_{n_1,n_2} \mathbb{E}_c \mathbb{E}_{\substack{i,j \\ (i,j) \in \Omega^{e,e'}}} \left[ \mathcal{D}(\phi_\Theta(A^e, X^e)_{[c][v_i]}, \phi_\Theta(A^e, X^{e'})_{[c][v_j]}) \right]$$
$$= \mathbb{E}_{\substack{e,e' \\ e \neq e'}} \mathbb{E}_{n_1,n_2} \mathbb{E}_c \mathbb{E}_{\substack{i,j \\ (i,j) \in \Omega^{e,e'}}} \| C_1^i + Z^e - C_1^j - Z^{e'} \|_2^2 \tag{74}$$

$$\frac{\partial \mathcal{L}_{\text{CIA}}}{\partial \theta_1} = \mathbb{E}_{\substack{e,e' \\ e \neq e'}} \mathbb{E}_{n_1,n_2} \mathbb{E}_c \mathbb{E}_{\substack{i,j \\ (i,j) \in \Omega^{e,e'}}} \left[ C_1^i\theta_1 + Z^e\theta_2 - C_1^j\theta_1 - Z^{e'}\theta_2 \right]^\top (C_1^i - C_1^j) \tag{75}$$

Let $\frac{\partial \mathcal{L}_{\text{CIA}}}{\partial \theta_1} = 0$, we have:

$$\mathbb{E}_{\substack{e,e' \\ e \neq e'}} \mathbb{E}_c \mathbb{E}_{\substack{i,j \\ (i,j) \in \Omega^{e,e'}}} \left[ (C_1^i - C_1^j)^\top (C_1^i - C_1^j)\theta_1 \right] = 0 \tag{76}$$

Thus, we get two possible solutions of the invariant branch. The first valid solution is the optimal one:

$$\begin{cases} \theta_1 = 1 \\ \theta_1^{1(l)} = 1, \theta_1^{2(l)} = 1, \quad l = L-1, ..., L-k+1 \\ \theta_1^{1(l)} = 0, \theta_1^{2(l)} = 1, \quad l = L-k, L-k-1, ..., 1 \end{cases}. \tag{77}$$

The second valid solution is a trivial one:

$$\left\{ \theta_1 = 0 \quad \text{or} \quad \exists l \in \{1, ..., L-1\} \text{ s.t. } \theta_1^{1(l)} = \theta_1^{2(l)} = 0 \right\}. \tag{78}$$

Take the derivative of the objective w.r.t. $\theta_2$:

$$\frac{\partial \mathcal{L}_{\text{CIA}}}{\partial \theta_2} = \mathbb{E}_{\substack{e,e' \\ e \neq e'}} \mathbb{E}_c \mathbb{E}_{\substack{i,j \\ (i,j) \in \Omega^{e,e'}}} \left[ \left[ (Z^e - Z^{e'})^\top (Z^e - Z^{e'}) \right] \theta_2 \right] = 2(1 + \sigma^2)\theta_2 \tag{79}$$

Let $\frac{\partial \mathcal{L}_{\text{CIA}}}{\partial \theta_2} = 0$, we get $\theta_2 = 0$. Thus, CIA will remove spurious features.

Now we show that when CIA objective has been reached (the spurious branch has zero outputs), the objective of $\min_\Theta \mathbb{E}_e[\mathcal{L}(f_\Theta(A^e, X^e), Y^e)]$ will help to learn predictive parameters of the invariant branch $\theta_1$, $\theta_1^{1(l)}$ and $\theta_1^{2(l)}$. When $\theta_2 = 0$:

$$
\begin{aligned}
\frac{\partial \mathbb{E}_e[\mathcal{L}(f_\Theta(A^e, X^e), Y^e)]}{\partial \theta_1} &= 2\mathbb{E}_e \mathbb{E}_{n_1, n_2} \left[ \left( C_1^e \theta_1 - \tilde{A}^{e^k} X_1 - n_1 \right)^\top C_1^e \right] \\
&= 2\mathbb{E}_e \left[ \left( C_1^e \theta_1 - \tilde{A}^{e^k} X_1 \right)^\top C_1^e \right]
\end{aligned}
\tag{80}
$$

Let $\frac{\partial \mathbb{E}_e[\mathcal{L}(f_\Theta(A^e, X^e), Y^e)]}{\partial \theta_1} = 0$, we get the predictive parameters

$$
\begin{cases}
\theta_1 = 1 \\
\theta_1^{1(l)} = 1, \theta_1^{2(l)} = 1, & l = L - 1, ..., L - k + 1 \\
\theta_1^{1(l)} = 0, \theta_1^{2(l)} = 1, & l = L - k, L - k - 1, ..., 1
\end{cases}
\tag{81}
$$

Plug the final solution back in $\frac{\partial \mathcal{L}_{\text{CIA}}}{\partial \theta_1^{1(l)}}$, $\frac{\partial \mathcal{L}_{\text{CIA}}}{\partial \theta_1^{2(l)}}$, $\frac{\partial \mathcal{L}_{\text{CIA}}}{\partial \theta_2^{1(l)}}$, $\frac{\partial \mathcal{L}_{\text{CIA}}}{\partial \theta_2^{2(l)}}$, $\frac{\partial \mathbb{E}_e[\mathcal{L}(f_\Theta(A^e, X^e), Y^e)]}{\partial \theta_1^{1(l)}}$, $\frac{\partial \mathbb{E}_e[\mathcal{L}(f_\Theta(A^e, X^e), Y^e)]}{\partial \theta_1^{2(l)}}$, $\frac{\partial \mathbb{E}_e[\mathcal{L}(f_\Theta(A^e, X^e), Y^e)]}{\partial \theta_2^{1(l)}}$, $\frac{\partial \mathbb{E}_e[\mathcal{L}(f_\Theta(A^e, X^e), Y^e)]}{\partial \theta_2^{2(l)}}$, we can verify that these terms are all 0. $\qquad\square$

