# OpenReview forum: "Improved Invariant Learning for Node-level Out-of-distribution Generalization on Graphs"
_ICLR.cc/2024/Conference — Submitted to ICLR 2024_

### Official Review · Reviewer_xLkZ · 2023-10-30

**Soundness:** 2 fair
**Presentation:** 3 good
**Contribution:** 2 fair
**Rating:** 5
**Confidence:** 4

**Summary:**

The paper presents a method called LoRe-CIA for out-of-distribution (OOD) generalization in graph neural networks (GNNs). The notable contributions of the paper include proposing a novel regularization term called LoRe, which encourages the learning of invariant representations across different environments. The LoRe term is combined with the Contrastive Invariant Alignment (CIA) objective to improve OOD generalization performance. Experimental results on the GOOD benchmark dataset demonstrate that LoRe-CIA outperforms several state-of-the-art methods in terms of OOD generalization accuracy.

**Strengths:**

The paper innovatively focuses on not fully explored node-level OOD generalization in graphs, revealing limitations of existing methods and introducing a novel approach, LoRe-CIA. High-quality work grounded in both theory. Addresses key challenges in graph-based OOD generalization and offers a robust solution with LoRe-CIA. Well-structured and clearly articulated, the paper provides a logical flow from problem formulation to solution, enhancing readability and comprehension.

**Weaknesses:**

The paper does not discuss the computational complexity of LoRe-CIA. Given that graph neural networks can be computationally intensive, an analysis of the algorithm's scalability to larger graphs would be beneficial. It would be better to add more recent models in experiments.

**Questions:**

In Table 2, when examining LoRe-CAI on the CBAS dataset, is there a specific reason why the variance of the results appears to be zero?

---

> ### Author Response · Authors · 2023-11-20
> **Response to Reviewer xLkZ**
>
> Thank you for your review!
>
> 1. **Time cost & complexity analysis**
>
>    To show the running time of LoRe-CIA, we run experiments on Arxiv (which has 50k~60k nodes). By our localized reweighting strategy, the time cost is controlled in a tolerable range.
>
>    Table A: (seconds per epoch) on Arxiv.
>
>    |                        | ERM  | IRM  | VREx | GroupDRO | Coral | DANN | Mixup | EERM | SRGNN  | CIA   | LoRe-CIA |
>    | ---------------------- | ---- | ---- | ---- | -------- | ----- | ---- | ----- | ---- | ------ | ----- | -------- |
>    | Arxiv degree covariate | 2.74 | 3.80 | 2.91 | 3.60     | 3.64  | 3.30 | 3.93  | OOM  | 141.29 | 30.82 | 10.35    |
>    | Arxiv degree concept   | 2.62 | 3.55 | 2.96 | 3.32     | 3.12  | 3.32 | 3.85  | OOM  | 130.64 | 7.23  | 12.72    |
>    | Arxiv time covariate   | 3.35 | 3.84 | 3.11 | 3.55     | 3.70  | 3.28 | 3.92  | OOM  | 65.66  | 27.53 | 14.03    |
>    | Arxiv time concept     | 3.17 | 3.46 | 2.89 | 2.88     | 3.56  | 3.23 | 3.81  | OOM  | 137.26 | 6.97  | 12.70    |
>
>    On smaller datasets, the running time gap is small and can be omitted. **The main time cost is the iteration over all classes.** For each class, we compute the difference of neighboring classes and compute L2 distances for all point pairs within K hop of each class. The theoretical time complexity upper bound of this procedure is $\mathcal{O}(C|E|)$, where $C$ is the number of classes and $|E|$ number of edges in a graph. However, since we only select the point pairs within $t$ hops and $t$ is relatively small (less than 5), the time cost is actually far less than  $\mathcal{O}(C|E|)$.
>
> 2. **Experiments on Recent Models**
>
>    Due to the tight timeline of rebuttal, we could only further test the performance on the off-the-shelf GIN network from the GOOD codebase. Results below show that LoRe-CIA offers significant improvements over other invariant learning methods, and CIA serves as a better invariant learning objective comparing to IRM and VREx.
>
>    Table B: OOD accuracy on Cora Using GIN.
>
>    | Cora                | dgree cov       | degree con      | word cov        | word con        |
>    | ------------------- | --------------- | --------------- | --------------- | --------------- |
>    | ERM                 | 52.30(0.19)     | 54.09(0.35)     | 62.35(0.01)     | 62.53(0.17)     |
>    | IRM                 | 52.23(0.44)     | 54.07(0.55)     | 62.50(0.23)     | 62.83(0.30)     |
>    | VREx                | 52.50(0.41)     | 54.56(0.14)     | 62.19(0.24)     | 62.56(0.35)     |
>    | CIA                 | 52.30(0.37)     | 54.54(0.48)     | 62.84(0.33)     | 62.92(0.44)     |
>    | LoRe-CIA 2ep starts | **52.76(0.28)** | **54.86(0.34)** | **62.92(0.38)** | **63.02(0.36)** |
>
> 3. **The zero std error problem:** this may be because CBAS covariate shift only has 70 test nodes, hence there is a high probability that the acc. takes several specific values, and it happens that 57 out of 70 nodes are predicted correctly.

---

### Official Review · Reviewer_q5yH · 2023-10-31

**Soundness:** 2 fair
**Presentation:** 2 fair
**Contribution:** 2 fair
**Rating:** 5
**Confidence:** 4

**Summary:**

This paper studies the out-of-distribution node classification. The authors propose a toy model considering the connections among nodes to demonstrate the limitations of IRMv1 and VREx in learning the underlying invariant features, such that similar frameworks like EERM well as fail. Then the authors incorporate the causal objecting matching objective to contrast intra-class samples across environments, called Cross-environment Intra-class Alignment (CIA). To address the collapse issue of the learned representations, the authors further propose a localized weighting variant of CIGA, called LoRe-CIA, that reweights the contrasting pairs according to the shortest path distance. In experiments with GOOD datasets, LoRe-CIA that does not use environment partitions, demonstrates competitive performance with respect to other methods that uses environment partitions.

**Strengths:**

(+) The studied problem is important;

(+) The theoretical failure results of IRM and VREx is interesting;

(+) The proposed localized reweighting scheme is new and interesting;

**Weaknesses:**

(-) The main motivation setting seems to be too hypothetical;

(-) The novelty of the proposed method is limited;

(-) The improvements are limited;

(-) Some related works have not been fairly discussed;

**Questions:**

Although I like the idea of the paper, it seems the paper could be improved significantly by clarifying the following points:

1. The main motivation setting seems to be too hypothetical.
- Although the proposed toy example seems to be more general than EERM, what are the realistic examples for the proposed data model? Without proper explanation and discussion, it is hard to claim that “This is the general case of real-world OOD graphs.”
- What does the superscript $e$ means in $A^e$? How environment changes could affect the generation of X?
- Moreover, the failure proofs seem to heavily rely on the assumption that “GNNs that are sufficiently deep (deeper than the number of layers of the causal pattern of a node) are typically used.”. Is it really true?

2. The novelty of the proposed method is limited and some related works have not been fairly discussed.
- The proposed method is a simple modification of Mahajan et al. (2021). How does the modification work to make up the environment information?
- If it’s only for the collapse issue, as already demonstrated in Figure 1, properly tuning the hyperparameter $\lambda$ already solves the problem well. How the $\lambda$ is tuned for CIA?
- What’s the relation between LoRe-CIA and CIA? Does CIA serve as a upper bound for LoRe-CIA? Why not incorporating LoRe into CIA with environments directly to resolve the collapse issue?
- What is the exact implementation of LoRe-CIA?
- The failures of IRM and VRex, and the success of intra-class contrastive learning are not surprising, as they are already shown by Chen et al., (2022). It is desirable to properly discuss the distinctions of the work with respect to previous works, thus the readers could better understand the place of the work in the literature.

3. The improvements are limited:
- When comparing to baselines that do not use environments, LoRe-CIA does not show clear advantage, and most of the improvements are within standard deviation.
- It is unclear how the hyperparameters such as $\lambda$ and hop numbers are tuned.

---

> ### Author Response · Authors · 2023-11-20
> **Response to Reviewer q5yH (part1)**
>
> - **"Although the proposed toy example seems to be more general than EERM, what are the realistic examples for the proposed data model? Without proper explanation and discussion, it is hard to claim that “This is the general case of real-world OOD graphs."**
>
>   Thank you for your careful review! We will first slightly extend our theoretical model to a more general version, in which we've changed the generation depth of the spurious features from $k$ to $m$,  i.e. we don't necessarily require the causal and spurious generation depth to be equal, while the main results still holds.
>
>   **Changes of our model:**
>
>   (previous) $ Y^e=\tilde{A^e}^kX_1+n_1,\quad X_2^e=\tilde{A^e}^kY^e+n_2+\epsilon^e=\tilde{A^e}^{2k}X_1+\tilde{A^e}^kn_1+n_2+\epsilon^e$, assume $L\geq2k$
>
>   (new) $ Y^e=\tilde{A^e}^kX_1+n_1,\quad X_2^e=\tilde{A^e}^mY^e+n_2+\epsilon^e=\tilde{A^e}^{k+m}X_1+\tilde{A^e}^mn_1+n_2+\epsilon^e$, assume $L\geq k$
>
>   **Realistic examples & further explanations:**
>
>   1. **Example 1: concept shift.** When m>0, our model represents the concept shift setting, i.e. $p(Y|X)$ varies with environment.  Given an environment $e$, the expectation of each dimension (each node) of $\epsilon^e$ is $\mu^e\in \mathbb{R}$, and different $Y$ of each node will produce different spurious features $X_2^e$. For example, for a citation network like Arxiv or Cora, given a year ($e$), different types of articles ($Y^e$) would tend to have different numbers of words ($X_2^e$). When given another year ($e'$), different types of articles ($Y^{e'}$) would still have different numbers of words ($X_2^{e'}$), and also the words of a certain type of articles will be different if the year varies, due to the environmental factor $\epsilon^e$.
>
>   2. **Example 2: covariate shift.** Our model can also be extend to covariate shifts where $p(X)$ changes across environments by removing the correlation between $X_2^e$ and $Y^e$, namely, $X_2^e=n_2+\epsilon^e$. **We add this part of analysis in Appendix A.** The theoretical results are the same the concept shift case: VREx and IRMv1 could use spurious features, while CIA lead to the optimal invariant solution. For the real-world example, consider WebKB dataset, where each node is a webpage with words appearing in the webpage as node features and edges are hyperlinks between webpages. Different universities ($e$) can  have webpages with different irrelevant features like style, ($\epsilon^e\rightarrow X_2^e$), while the categories of the webpages just have to with word contents.

---

> ### Author Response · Authors · 2023-11-20
> **Response to Reviewer q5yH (part2)**
>
> 3. **Why multi-layer data generation process is needed.** The multi-layer generation depth corresponds to the real case that causal/spurious features of a node might be affected by more than 1-hop neighbors. We empirically verify this on Arxiv and Cora. "causal"/"spurious" means we aim to predict $Y$ or $e$. Since a GCN with layer $l$ will aggregate features from $l$-hop neighbors for prediction, if the depth of the GCN is equal to the true generation depth, then the performance should be close to optimal.  The Table A,B,C,D below indicate **the causal/spurious features for a node can be determined by neighbors exceed one hop, hence our modeling of multi-layer feature generation process is necessary.**
>
>      Table A: optimal numbers of layers on Cora (predict ground-truth label $Y$):
>
>      | layers of GCN         | 1               | 2               | 3           | 4           |
>      | --------------------- | --------------- | --------------- | ----------- | ----------- |
>      | Cora degree covariate | **59.04(0.15)** | 58.44(0.44)     | 55.78(0.52) | 55.15(0.24) |
>      | Cora degree concept   | **62.88(0.34)** | 61.53(0.48)     | 60.24(0.40) | 60.51(0.17) |
>      | Cora word covariate   | 64.05(0.18)     | **65.81(0.12)** | 65.07(0.52) | 64.58(0.10) |
>      | Cora word concept     | 64.76(0.91)     | **64.85(0.10)** | 64.61(0.11) | 64.16(0.23) |
>
>      Table B: optimal numbers of layers on Cora (predict environment label $e$):
>
>      | layers of GCN         | 1               | 2           | 3           | 4           |
>      | --------------------- | --------------- | ----------- | ----------- | ----------- |
>      | Cora degree covariate | **97.94(0.09)** | 89.41(0.85) | 88.24(0.51) | 84.12(0.51) |
>      | Cora degree concept   | **95.86(0.09)** | 92.63(0.29) | 91.34(0.48) | 89.84(0.25) |
>      | Cora word covariate   | **95.22(0.03)** | 91.98(0.14) | 90.20(0.06) | 84.79(0.49) |
>      | Cora word concept     | **95.81(0.07)** | 92.01(0.25) | 92.07(0.22) | 90.15(0.22) |
>
>      Table C: optimal numbers of layers on Arxiv (predict ground-truth label $Y$):
>
>      | layers of GCN          | 2           | 3           | 4               | 5               |
>      | ---------------------- | ----------- | ----------- | --------------- | --------------- |
>      | Arxiv degree covariate | 57.28(0.09) | 58.92(0.14) | **60.18(0.41)** | 60.17(0.12)     |
>      | Arxiv degree concept   | 63.32(0.19) | 62.92(0.21) | **65.41(0.13)** | 63.93(0.58)     |
>      | Arxiv time covariate   | 71.17(0.21) | 70.98(0.20) | **71.71(0.21)** | 70.84(0.11)     |
>      | Arxiv time concept     | 65.14(0.12) | 67.36(0.07) | 65.20(0.26)     | **67.49(0.05)** |
>
>      Table D: optimal numbers of layers on Arxiv (predict environment label $e$):
>
>      | layers of GCN          | 1           | 2               | 3               | 4           |
>      | ---------------------- | ----------- | --------------- | --------------- | ----------- |
>      | Arxiv degree covariate | 21.30(0.85) | **30.16(0.39)** | 26.16(0.70)     | 15.36(0.56) |
>      | Arxiv degree concept   | 43.10(0.08) | 46.92(0.49)     | **48.77(0.16)** | 47.98(0.26) |
>      | Arxiv time covariate   | 33.01(0.24) | 36.91(0.67)     | **44.26(0.24)** | 41.76(0.48) |
>      | Arxiv time concept     | 42.71(0.05) | 46.14(0.49)     | **48.03(0.14)** | 43.10(0.08) |

---

> ### Author Response · Authors · 2023-11-20
> **Response to Reviewer q5yH (part3)**
>
> - **What does the superscript e means in $A^e$? How environment changes could affect the generation of X?**
>
>   $A^e$ means the adjacency matrix of a subgraph from environment $e$. Change of environments can affect the generation of $X$ by two ways: (1) different  environments will have $\epsilon^e$  of different expectations (we assume $\mathbb{E}_{\epsilon_i\sim p_e}[\epsilon_i]=\mu^e\in \mathbb{R}$), thus leading to different distributions of spurious features $X_2^e$, (2) different environments will have different topological structures $A^e$, this will also produce different $X$.
>
> - **Moreover, the failure proofs seem to heavily rely on the assumption that “GNNs that are sufficiently deep (deeper than the number of layers of the causal pattern of a node) are typically used.”. Is it really true?**
>
>   In our extended model, we assume $L\geq k$. To empirically find out how large $k$ really is, we use GCN with different numbers of layers to predict the ground-truth label $Y$ on Cora and Arxiv (results are in Table A and B above). As mentioned above, since a GCN with layer $l$ will aggregate features from $l$-hop neighbors for prediction, if the depth of the GCN is equal to the true generation depth, then the performance should be close to optimal. Suppose the empirical optimal layer number is $L^*$ for prediction, we have: $L^*=k$.
>
>   **We find that the $L^*\leq 4$ in most cases (even on large-scale graphs in Arxiv).**  This indicates that our assumptions holds easily.  Detailed results and analysis are in Appendix D.3.
>
> - **The proposed method is a simple modification of Mahajan et al. (2021). How does the modification work to make up the environment information?**
>
>   The connection of nodes plays a role of auxillary information for us to make up the absence of environment information. Intuitively, since environment labels play the role of filtering nodes pairs with larger differences in spurious features and smaller differences in invariant features in  CIA, it is possible that we could achieve the same invariant learning goal as long as we could find such node pairs. This is the main idea of LoRe-CIA. It uses neighboring label distribution to achieve this. We also add additional experiments to show that neighboring label distribution can reflect the invariant/spurious feature distribution of a centered node, please see the next reply.
>
> - **If it’s only for the collapse issue, as already demonstrated in Figure 1, properly tuning the hyperparameter $\lambda$ already solves the problem well. How the $\lambda$ is tuned for CIA?**
>
>   For all hyperparameters in this paper, we use grid search in a fixed seach space, please see Appendix B.2 for hyperparamter settings.
>
> - **What’s the relation between LoRe-CIA and CIA?**
>
>   LoRe-CIA serves as an alternative when environment label is not available. Also, LoRe-CIA has less time and space complexity but better empirical performances than CIA.
>
> - **Does CIA serve as a upper bound for LoRe-CIA?**
>
>   We didn't prove that CIA serves as a generalization performance upper bound for LoRe-CIA. However, we show in our experiments that LoRe-CIA can outperform CIA even without environment labels. This proves the effectiveness of our strategy for selecting node pairs.  In fact, **we add experiments in Appendix C.4 and C.5, and empirically prove the correctness of the intuition of LoRe-CIA:** (1) the change rate of node’s spurious features w.r.t spatial location is faster than that of the causal/invariant features within a certain range of hops (about 5~10 hops), thus localized alignment can better eliminate spurious features and keep the diversity of causal features; (2) there is a clear positive correlation between the spurious feature distance and class-different neighboring label discrepancy.
>
>   We think your question about the relationship between theoretical performance of CIA and LoRe-CIA is interesting and we would like to leave it for our future exploration.
>
> - **Why not incorporating LoRe into CIA with environments directly to resolve the collapse issue?**
>
>   As stated in our paper, environment labels are often hard to acquire. Therefore, the main aim of this paper is to find an effective alternative to compensate for the lack of environmental labeling, meanwhile leveraging the advantage of CIA as predicted by the theory.
>
> - **What is the exact implementation of LoRe-CIA?**
>
>   We have posted the pseudo code in Appendix D. We'll release the source code after the publication of this paper.

---

> ### Author Response · Authors · 2023-11-20
> **Response to Reviewer q5yH (part4)**
>
> - **The failures of IRM and VRex, and the success of intra-class contrastive learning are not surprising, as they are already shown by Chen et al., (2022). It is desirable to properly discuss the distinctions of the work with respect to previous works, thus the readers could better understand the place of the work in the literature.**
>
>   We're sorry that we have not been able to locate the article you are referring to. We would be very grateful if you could provide the title of "Chen et al., (2022).".
>
>   To the best of our knowledge, we are the first work to theoretically analysis the limitations of IRMv1 and VREx in OOD node classification problems. Especially, the VREx objective has been adopted by mant graph OOD methds  (EERM, LiSA, DIR, G-Splice, GIL).  Moreover, we are the first work to apply CIA on node-level OOD tasks and theoretically/emipirically reveals its superiority. We hope to throw light on this better invariant learning objective for future node-level OOD tasks by this paper. Additionally, we emipirically reveal that even without environment labels, with appropriate use of graph structure and neighbor information, we can achieve remarkable empirical generalization results. This observation motivates future works to  For further comparison between our work and previous works, we'd like to invite you to refer to focus more on how to use the graph-specific information, such as neighboring features and local/global structures to enhance the generalization on graphs. Related Work Section (Section 6).
>
> - **When comparing to baselines that do not use environments, LoRe-CIA does not show clear advantage, and most of the improvements are within standard deviation.**
>
>   As obeserved in GOOD: A Graph Out-of-Distribution Benchmark (Gui et al, 2022), it is hardly possible for any algorithm to consistently ouperform other methods even including ERM on all splits. For example, in the experiments in (Gui et al, 2022) even though Mixup (Wang et al, 2021) has the best performance on node-level OOD tasks, **it merely leads 6 out of 14 node-task OOD splits.** In our experiments (along with the new added experiments on large-scale Arxiv dataset, included in Table 2&3 in the main text), **LoRe-CIA leads 7 out of 12 splits and has the best average performance, Mixup leads 4 out of 12 splits and CIA leads 1 split** (note that we use the same codebase adopted from GOOD benchmark, so the results are fair). Further, the improvements of the leading algorithms in the experiments presented in (Gui et al, 2022) are also within standard deviation in many cases (you could refer to the Table 15~21 in (Gui et al, 2022)). Generally, we think it reasonable to consider LoRe-CIA as an effective solution when environment labels are absent for OOD node classification.
>
>   Additionally, comparing to other methods without using environment like SRGNN, Mixup and EERM, the performance of LoRe-CIA is stable across real-world and synthetic datasets on all splits. For example, SRGNN Mixup and EERM all show severe performance degradation on CBAS while LoRe-CIA remains its good performance.
>
> - **It is unclear how the hyperparameters such as $\lambda$ and hop numbers are tuned.**
>
>   For all hyperparameters in this paper, we use grid search in a fixed seach space, please see Appendix B.2 for hyperparamter settings.

---

> > ### Comment · Reviewer_q5yH · 2023-11-22
> >
> > Thank you for the detailed explanations, which have addressed some of my concerns. Nevertheless, many of my questions remain unanswered:
> > - In part 1 and part 2, the authors explained a bit about their theoretical example, while it remains unknown for the necessity of the data model, especially compared to the previous one established in EERM. It seems the key difference between the two models is the multi-layer structure, which seems to be unnecessary for the two real cases given by the authors;
> > - "if the depth of the GCN is equal to the true generation depth, then the performance should be close to optimal" seems to be a vague claim. For a real-world graph, many factors could affect the performance. For example, for a graph with high homophily, a deeper GNN could have a better performance, while for a heterophilous graph, a simple MLP usually can have a better performance than any layers of GNNs.
> > - From the authors' explanation of how LoRe-CIA works, it seems LoRe-CIA takes a "free lunch" that could improve over the original variant CIA (Mahajan et al., 2021) with environment information, by incorporating the node pair information. It would be better if the authors could justify their approach in a rigorous form, which could be helpful to better understand why LoRe-CIA could outperform and underperform previous approaches and further improve LoRe-CIA.
> >
> > Given the aforementioned reasons, I have to maintain my original rating of this work for now.

---

> > > ### Author Response · Authors · 2023-11-23
> > > **Further response to Reviewer q5yH**
> > >
> > > - **In part 1 and part 2, the authors explained a bit about their theoretical example, while it remains unknown for the necessity of the data model, especially compared to the previous one established in EERM. It seems the key difference between the two models is the multi-layer structure, which seems to be unnecessary for the two real cases given by the authors;**
> > >
> > >   Sorry to keep you in doubt. We show the necessity of the multi-layer data generation in the part 2 of the rebuttal, the results of Table C and D indicate that the groud-truth label of a node is not only determined by 1-hop neighbors since when the accuracy increases with the number of GNN layers. We believe the multi-layer data generation is more reasonable and general for real-world graphs since we cannot assume the label of a node  is solely determined by its 1-hop neighbors. Further, our multi-layer model reveals the failure of VREx which was not shown in the 1-layer case in EERM, which highlights the non-triviality of the multi-layer setting.
> > >
> > > - **"if the depth of the GCN is equal to the true generation depth, then the performance should be close to optimal" seems to be a vague claim. For a real-world graph, many factors could affect the performance. For example, for a graph with high homophily, a deeper GNN could have a better performance, while for a heterophilous graph, a simple MLP usually can have a better performance than any layers of GNNs.**
> > >
> > >   We agree with the finding of " for a graph with high homophily, a deeper GNN could have a better performance, while for a heterophilous graph, a simple MLP usually can have a better performance than any layers of GNNs". The aim of our $L\geq k$ assumption is to ensure that the GNNs have enough capability to predict correctly, since in our toy model, $Y=A^kX_1+n_1$, if $L<k$ then no objective will lead to a solution of GNN parameters that can predict $Y$ accurately. Our $L\geq k$ assumption is just used for simplifying the analysis.  **We actually don't require that all information that is predictive for a node's label is within $k$-hop, we just require that information within $k$-hop is enough for produce a relvatively accurate prediction. In other words, as long as the GNN is deep enough to reach its best performance it can reach, our theoretical results will hold.** For heterohilous graphs, $L\geq k $ can be satisfied easily since $k=0$ in many real cases (see Figure 8 of ADAPTIVE UNIVERSAL GENERALIZED PAGERANK GRAPH NEURAL NETWORK ( Chien & Peng et al, ICLR 2021) ). For homophilous graphs (like Arxiv and Cora), we have evaluated in part 2 that four layers is enough for a GNN to reach its best performance.
> > >
> > > - **From the authors' explanation of how LoRe-CIA works, it seems LoRe-CIA takes a "free lunch" that could improve over the original variant CIA (Mahajan et al., 2021) with environment information, by incorporating the node pair information. It would be better if the authors could justify their approach in a rigorous form, which could be helpful to better understand why LoRe-CIA could outperform and underperform previous approaches and further improve LoRe-CIA.**
> > >
> > >   Thank you for your advice! Using the theoretical framework we propose, we can prove that LoRe-CIA can remove spurious features given the assumption that the differences in spurious/causal features grow mononotically with the differences in neighboring same/different-class label distribution (which is empirically observed in Appendix C.5). However, theoretically analyzing the role of LoRe-CIA in solving causal feature collapse needs reformulation of the current theoretical model, which is complicated and needs careful exploration. We will leave this part in future work.

---

> ### Comment · Reviewer_q5yH · 2023-11-23
>
> Thank you for the follow-up explanation.
>
> - From the explanation, it is still hard to find specific reasons for adopting the multi-layer model. From the theoretical proving perspective to demonstrate that VREx can not address the issue, the setup is reasonable, while the experiments can not justify the necessity of the multi-layer model. In fact, in EERM, the label is generated conditioned on the set of nodes of an ego-graph, which takes account of multiple hops too.
>
> - As acknowledged by the authors, there are multiple simplified theoretical setups for the ease of analysis, yet the simplicity seems not to be sufficiently justified, especially from the perspective of real-world practice.
>
> Although the previous responses have addressed some of my concerns, given the limited theoretical and methodology novelty, and the empirical performance, I have slightly increased the rating but only to 5.

---

### Official Review · Reviewer_vEQV · 2023-10-31

**Soundness:** 3 good
**Presentation:** 3 good
**Contribution:** 2 fair
**Rating:** 5
**Confidence:** 4

**Summary:**

The paper studies node-level OOD generalization. It establish cases where invariant learning methods like IRM and VREx given environment labels learn spurious features when the depth of the GNN exceeds the causal depth. Then it shows the cross-environment intra-class alignment of nodes avoid some spurious features. The paper also proposes LoRe-CIA, which does not require environment labels but selects node pairs that exhibit large differences in spurious features but minimal differences in causal features for alignment. Experiments are conducted to evaluate the methods.

**Strengths:**

1. The paper studies an algorithmic case and clearly analyzes IRM, VREx and CIA from a theoretical perspective.

2. The paper presentation is clear and easy to follow.

3. The proposed method does not require ground-truth environment labels. The conducted experiments look correct.

**Weaknesses:**

1.  The setting in 3.1 assumes $L > 2k$ and GNNs are sufficiently deep, which is doubtful. Typically graph tasks would not use very deep GNNs due to the well-known fact that deep GNNs have over-smoothing issues and compromise performances. Therefore this assumption may not hold. Moreover, there's no validation for the number of layers of the causal pattern of a node in any real-world dataset, thus no conclusion whether this $L > 2k$ setting is applicable for any graph tasks.

2. From my understanding, CIA's major superiority over other methods using environment labels like VREx roots in its consideration of both class and environment label information at the same time. However, LoRe-CIA does not use environment labels. According to [1], learning invariant/spurious features without environment partition is fundamentally impossible if not given further inductive biases or information. Thus, there appears to be no guarantee that LoRe-CIA can learn the information supporting CIA, i.e., LoRe-CIA might not be able to identify node pairs with significant differences in spurious features and small differences in causal features.

3. Sec 4.2 assumes 1. change rate of node’s spurious features w.r.t spatial location is faster than that of the causal feature; 2. label distribution of node's different-class neighbors reflects the distribution of spurious features. This assumption is quite strong and seems not applicable widely. There are graphs exhibiting various behavior, such as heterogeneous/homogeneous graphs. Further discussions on when these assumptions hold should be included.


[1] ZIN: When and How to Learn Invariance Without Environment Partition?

**Questions:**

See weaknesses

---

> ### Author Response · Authors · 2023-11-20
> **Response to Reviewer vEQV (part1)**
>
> $L\geq 2k$ assumption:
>
> Thank you for your careful reading! We will first slightly extend our theoretical model to a more general version, and then we will show how to empirically validate the generation depth of causal features. Finally we'll show the experimental evidence supporting this assumption.
>
> **Changes of our model:**
>
> We sightly improve our theoretical model to use weaker assumption, while the main conclusions remain unchanged:
>
> (previous) $ Y^e=\tilde{A^e}^kX_1+n_1,\quad X_2^e=\tilde{A^e}^kY^e+n_2+\epsilon=\tilde{A^e}^{2k}X_1+\tilde{A^e}^kn_1+n_2+\epsilon$, assume $L\geq 2k$
>
> (new) $$ Y^e=\tilde{A^e}^kX_1+n_1,\quad X_2^e=\tilde{A^e}^mY^e+n_2+\epsilon=\tilde{A^e}^{k+m}X_1+\tilde{A^e}^mn_1+n_2+\epsilon$$, assume $L\geq k$
>
> We've changed the generation depth of the spurious features from $k$ to $m$,  i.e. we don't necessarily require the causal and spurious generation depth to be equal, while the main results still holds (with onlt slight changes in the proof). Now, we only require $L\geq k$ . All our theoretical results still hold.
>
> To validate the depth of the causal generation pattern, we use GCN with different layer to predict the ground-truth label $Y$ on Cora and Arxiv dataset respectively (results are in Table A and B below). Since a GCN with layer $l$ will aggregate features from $l$-hop neighbors for prediction, if the depth of the GCN is equal to the true generation depth, then the performance should be close to optimal. Suppose the empirical optimal layer number is $L^*$ for prediction, we should have: $L^*=k$.
>
> The optimal number of layers of GNN is listed below in Table A and B.  **We find that the $L^*\leq4$ in most cases (even on large-scale graphs in Arxiv).** And GNNs of 4~5 layers are commenly used. This indicates that our assumptions holds easily.  Detailed results and analysis are in Appendix D.3 in the paper.
>
> Table A. optimal number of layers on Cora
>
> | layers of GCN         | 1               | 2               | 3           | 4           |
> | --------------------- | --------------- | --------------- | ----------- | ----------- |
> | Cora degree covariate | **59.04(0.15)** | 58.44(0.44)     | 55.78(0.52) | 55.15(0.24) |
> | Cora degree concept   | **62.88(0.34)** | 61.53(0.48)     | 60.24(0.40) | 60.51(0.17) |
> | Cora word covariate   | 64.05(0.18)     | **65.81(0.12)** | 65.07(0.52) | 64.58(0.10) |
> | Cora word concept     | 64.76(0.91)     | **64.85(0.10)** | 64.61(0.11) | 64.16(0.23) |
>
> Table B. optimal number of layers on Arxiv
>
> | layers of GCN          | 2           | 3           | 4               | 5               |
> | ---------------------- | ----------- | ----------- | --------------- | --------------- |
> | Arxiv degree covariate | 57.28(0.09) | 58.92(0.14) | **60.18(0.41)** | 60.17(0.12)     |
> | Arxiv degree concept   | 63.32(0.19) | 62.92(0.21) | **65.41(0.13)** | 63.93(0.58)     |
> | Arxiv time covariate   | 71.17(0.21) | 70.98(0.20) | **71.71(0.21)** | 70.84(0.11)     |
> | Arxiv time concept     | 65.14(0.12) | 67.36(0.07) | 65.20(0.26)     | **67.49(0.05)** |

---

> ### Author Response · Authors · 2023-11-20
> **Response to Reviewer vEQV (part2)**
>
> 1. "LoRe-CIA might not be able to identify node pairs with significant differences in spurious features and small differences in causal features"
>
>    We agree with the opinions in [1] that learning invariant/spurious features without environment partition is fundamentally impossible if not given further inductive biases or information. However, **the "further inductive biases or information" in LoRe-CIA is the connectivity of the nodes and the neighboring label distribution.**  The point you mentioned that "consideration of both class and environment label information at the same time" is indeed one of the superiorities of CIA. Also, more important is the way it leverages environment labels: environment labels play the role of filtering nodes pairs with larger differences in spurious features and smaller differences in invariant features, and this goal can be reached alternatively by LoRe-CIA objective that uses auxilliary information from neighboring samples. **We add additional experiments to show that neighboring same-class/different-class label distribution can reflect the invariant/spurious feature distribution of a centered node (in Appendix C.5.1 and C.5.2).**
>
>
>
> 2. Hypothesis 1"change rate of node’s spurious features w.r.t spatial location is faster than that of the causal feature":
>
>    We conduct expriments to evaluate the change rate of the invariant features and spurious features w.r.t. spatial position, and found that **the changes of spurious features grow faster w.r.t. the change of spatial position than the causal feature within certain range of hops (about 5~10 hops, and the range of the hop hyperparameter adopted in LoRe-CIA is [2,5])**. Therefore, the strategy of local alignment can well avoid collapse of causal features and better eliminate spurious features. **Experimental results supporting this hypothesis are listed in Appendix C.4.**
>
>    Hypothesis 2"label distribution of node's different-class neighbors reflects the distribution of spurious features":
>
>    We also add additional expriments to verify this intuition in Appendix C.5.1, and empirically verified **a clear positive correlation between the spurious feature distance and Heterophilous Neighboring Label Distribution (HNLD) discrepancy, for both concept and covariate shift, both node feature shift datasets and graph structure shift datasets.** Here is the intuitive explaination for this. Spurious features of a node come from two sources: (1) the environmental spurious feature and (2) class-different (heterophilous) neighboring features.  For covariate shifts, since spurious features are not necessarily correlated with labels, the environmental spurious features cannot be reflected by HNLD. However, we can still measure the distribution of the spurious features caused by heterophilous neighbors. For concept shift, spurious features are correlated with labels, thus the label distribution contains information about spurious features correlated with this class.

---

> > ### Author Response · Authors · 2023-11-23
> > **Your comments are important for us**
> >
> > Dear Reviewer vEQV,
> >
> > There are only several hours left for the discussion. Hope we could have the last chance to discuss with you! We really have spent lots of time and effort preparing a very very detailed response. Your comments are very important to us. Could you please have a look? Many thanks!
> >
> > Have a nice day!

---

### Official Review · Reviewer_v4b8 · 2023-11-06

**Soundness:** 3 good
**Presentation:** 2 fair
**Contribution:** 3 good
**Rating:** 6
**Confidence:** 4

**Summary:**

This paper presents a novel method to improve out-of-distribution (OOD) generalization for node-level tasks on graph data. It introduces a theoretical model to assess OOD techniques in node-level OOD classification and proposes an enhanced invariant learning objective that considers both graph topology and node features. The method is evaluated on benchmark datasets, demonstrating superior OOD detection and classification performance. The paper contributes a theoretical analysis of OOD methods on graphs, an innovative approach for invariant learning, and empirical benchmark evaluations.

**Strengths:**

- **Motivation**: The paper addresses an important challenge in graph learning: achieving OOD generalization on real-world graphs.

- **Theoretical guarantee**: This approach is grounded in a theoretical model that analyzes the performance of several OOD methods, including V-REx and IRM, in node-level OOD classification problems. The theoretical model presented in the paper also offers insights into the performance of OOD methods, which could inform future research in this area.

- **Statement of intuition**: The paper provides explanations of the theoretical model and the proposed approach. I particularly appreciate the part where the authors discuss the intuitions presented in the main paper.

**Weaknesses:**

- **Theoretical model**: In Remark, the authors noted, "In this toy model, the distribution shift is caused by both changes in topological structures (Ae) and node features (Xe2). This represents the general case of real-world OOD graphs." It would be beneficial to see a more detailed analysis of the model's assumptions and limitations. Specifically, what cases are not covered by the model?

- **Efficiency explanation**: In the abstract and introduction, the authors mentioned that the proposed approach enables "more efficient elimination of spurious features." However, the paper does not provide an analysis of the computational complexity of this approach, and the experimental section lacks relevant information about its runtime.

- **Presentation**: The paper's structure and clarity could be further improved. For instance, in Figures 1 and 2, the shapes representing labels are hard to tell at the first sight. In Section 4.2, the statement "the rate of change of a node’s spurious features with respect to spatial location on the graph is faster than that of the causal feature" could benefit from a more specific explanation of what "rate" means in this context. Additionally, while the content is generally clear, the writing could be further refined.

- **Real-world graph datasets**: Since CBAS is a synthetic dataset, its weight in the study may be smaller than the other two datasets. It might be worthwhile to include more real-world datasets to validate the generalization ability of CIA/LoRe-CIA across a wider range of distributions.

**Questions:**

- What are the cases that are not included in the theoretical model?
- What is the evidence that supports the claim about CIA is a more efficient approach?
- In Section 4.2, "...the rate of change of a node’s spurious features with respect to spatial location on the graph is faster than that of the causal feature...", what does "rate" mean specifically?

---

> ### Author Response · Authors · 2023-11-20
> **Response to Reviewer v4b8 (part1)**
>
> 1. "What are the cases that are not included in the theoretical model?"
>
>    Thank you for your questions!  We assume the spurious feature of a node is  affected by the labels of the node itself and that of its neighbors (and this effection varies with environment through $\epsilon^e$): $X_2^e=\tilde{A^e}^kY^e+n_2+\epsilon^e$. This represents a concept shift setting where spurious features are correlated $Y$. Our original model didn't cover the case that $X_2^e$ is independent of $Y$, i.e. the covariate shift. **We have added this part of analysis in Appendix A in the paper, where we remove the dependent of $X_2^e$ on $Y$ and the distribution shift is caused solely by $\epsilon^e$ which varies with environments, and the main theoretical results remains the same as the concept shift case: VREx and IRMv1 could use spurious features and CIA achieves the optimal invariant solution.**
>
>    Additionally, our model presented in the main text  only considers a linear generation process of invariant and spurious features: $ Y^e=\tilde{A^e}^kX_1+n_1,\quad X_2^e=\tilde{A^e}^kY^e+n_2+\epsilon^e $, and there could be more complex non-linear data generation process in real-world scenarios. We will try including more general generation process in future works.
>
>
>
> 2. Efficiency explanation:
>
>    First of all, we need to apologize for our inaccurate express. Actually, "efficiency" in our paper means that LoRe-CIA selectively finds node pairs with larger spurious feature differences and smaller causal feature differences than CIA does, so that our method can achieve good generalization even without environment labels.  We have changed our expression in the pdf. Following your suggestion,  we run experiments on Arxiv (which has 50k~60k nodes) to show the running time of LoRe-CIA.
>
>    Table 1. (seconds per epoch) on Arxiv.
>
>    |                        | ERM  | IRM  | VREx | GroupDRO | Coral | DANN | Mixup | EERM | SRGNN  | CIA   | LoRe-CIA |
>    | ---------------------- | ---- | ---- | ---- | -------- | ----- | ---- | ----- | ---- | ------ | ----- | -------- |
>    | Arxiv degree covariate | 2.74 | 3.80 | 2.91 | 3.60     | 3.64  | 3.30 | 3.93  | OOM  | 141.29 | 30.82 | 10.35    |
>    | Arxiv degree concept   | 2.62 | 3.55 | 2.96 | 3.32     | 3.12  | 3.32 | 3.85  | OOM  | 130.64 | 7.23  | 12.72    |
>    | Arxiv time covariate   | 3.35 | 3.84 | 3.11 | 3.55     | 3.70  | 3.28 | 3.92  | OOM  | 65.66  | 27.53 | 14.03    |
>    | Arxiv time concept     | 3.17 | 3.46 | 2.89 | 2.88     | 3.56  | 3.23 | 3.81  | OOM  | 137.26 | 6.97  | 12.70    |
>
>    On smaller datasets, the running time gap is small and can be omitted, even on large datasets like Arxiv, the time cost of LoRe-CIA is tolerable. **The main time cost is the iteration over all classes.** For each class, we compute the difference of neighboring classes and compute L2 distances for all point pairs within K hop of each class. The theoretical time complexity upper bound of this procedure is $\mathcal{O}(C|E|)$, where $C$ is the number of classes and $|E|$ number of edges in a graph. However, since we only select the point pairs within $t$ hops and $t$ is relatively small (less than 5), the time cost is actually far less than  $\mathcal{O}(C|E|)$.
>
> 3. what does "rate" mean specifically?
>
>    We apologize for the confusion. Also, we're very sorry for the unrigorous statement we've given: the assumption we're relying on is actually "spurious feature changes faster w.r.t. the change of spatial position than the invariant feature **within a certain range of hops**.  The exact definition of the "rate" is as follows. Given any node in the graph, we can generate a series of BFS (Breadth-First Search) paths from this staring node. Then the change rate of the features can be reflected by calculating the distances of the representation between the nodes on the BFS path and the starting node.  **We've added experimental results on real-world datasets and discussions to support this hypothesis in Appendix C.4.**  **We empirically find that within about 5~10 hops (and the range of the hop hyperparameter adopted in LoRe-CIA is [2,5]), spurious features exhibit greater changes than invariant features.**  Therefore, the strategy of local alignment is adopted to avoid the collapse of the causal feature and the reliance on spurious features.

---

> ### Author Response · Authors · 2023-11-20
> **Response to Reviewer v4b8 (part2)**
>
> 1. Additional results on Arxiv (large-scale real-world dataset).
>
>    We add experiments on GOOD-Arxiv to further test the real-world generalization of our method. Arxiv contains 50~60k training nodes for each shift, and is larger than all datasets we listed in the main text. **Results show that LoRe-CIA shows stronger generalization ability than most previous methods, and also, CIA outperforms IRM and VREx as predicted by our theory.**
>
>    | Arxiv      | dgree cov       | degree con      | time cov        | time con        |
>    | ---------- | --------------- | --------------- | --------------- | --------------- |
>    | ERM        | 58.92(0.14)     | 62.92(0.21)     | 70.98(0.20)     | 67.36(0.07)     |
>    | IRM        | 58.93(0.17)     | 62.79(0.11)     | 70.86(0.12)     | 67.42(0.08)     |
>    | VREx       | 58.75(0.16)     | 63.06(0.43)     | 69.80(0.21)     | 67.42(0.07)     |
>    | EERM       | OOM             | OOM             | OOM             | OOM             |
>    | GroupDRO   | 58.87(0.00)     | 62.98(0.53)     | 70.93(0.09)     | 67.41(0.27)     |
>    | Deep Coral | 59.04(0.16)     | 63.09(0.28)     | 71.04(0.07)     | 67.43(0.24)     |
>    | DANN       | 59.03(0.15)     | 63.04(0.20)     | 71.09(0.03)     | 67.46(0.23)     |
>    | Mixup      | 57.80(0.19)     | 62.33(0.34)     | **71.62(0.11)** | 65.28(0.43)     |
>    | SRGNN      | 58.47(0.00)     | 62.80(0.25)     | 70.83(0.10)     | 67.17(0.23)     |
>    | CIA        | 59.03(0.39)     | 63.87(0.26)     |  71.10(0.15)         | **67.62(0.04)** |
>    | LoRe-CIA   | **59.12(0.18)** | **63.89(0.31)** | 71.16(0.11)     | 67.52(0.10) |
>
> 2. Presentation issue: thank you for your opinions! Due to limited rebuttal time, we're sorry that have not had time to modify the appreance of Figure 1 and the structure of the full text. We will carefully fix these issues in the time between the end of rebuttal and the release of the review results.

---

> ### Author Response · Authors · 2023-11-23
> **Comments on the rebuttal?**
>
> Dear Reviewer v4b8,
>
> We understand that perhaps you are too busy to read the rebuttal. But since there are only a few hours left, we are sorry to remind you again.
>
> In our response, we have addressed your concerns on the limitations of our theoretical model (which we have fixed), efficiency explainations, the definition of "change rate" (which we conduct further expriment to validate our key assumptions) and real-world dataset experiments (we conduct further experiments on Arxiv and prove the effectiveness of LoRe-CIA).
>
> We are wondering if our response satisfies you? We are happy to answer any further questions.
>
> Sincerely, Authors

---

> > ### Comment · Reviewer_v4b8 · 2023-11-23
> >
> > I acknowledge the aurthors' response, and apologize for the late response.
> >
> > The authors has addressed most of my concern, while I believe the presentation can be further refined judging from the current revision. For example, a causal graph can be added to the assumtion; it might not be necessary to discuss both IRM and VREx in the main paper - an in-depth discussion on one of them should be fine; it would also be nice to give more intuitive illustration on why they fail.
> >
> > I am willing to increase my score to 6.

---

### Meta-Review · Area_Chair_xhhr · 2023-12-10

**Metareview:**

This submission investigates the failures of popular OOD-robustness methods in OOD node classification tasks. I really like the idea of introducing a simple data generation process to explain why VRex and IRM can learn spurious correlations. The "linear" GNN introduced by the work is could have been a little more general but it is common to make this simplification in order to obtain theoretical results. The simplicity of the data generation model, however, is less justified. Just having node features in the graph being the driver of node labels is a little too restrictive. What about structure? Would VRex and IRM also fail under structural spurious associations? The proposed LoRe-CIA being tied to spurious features alone feels too narrow of an approach that could be expanded for a significant more impactful contribution.

After reading the reviewer-author discussions and the paper, I more positive than the average reviewer score. I think this is a solid borderline paper. To go beyond "borderline" into solid "accept" territory I feel the paper needs to consider simple spurious structural features, and adapt the theory and LoRe-CIA to these scenario. "Linear" GNNs are not that hard to incorporate structural features into the theory.

**Justification For Why Not Higher Score:**

The theory is somewhat basic with a too-simple data generation model. I like the concept a lot, but the theory needs to include structural features.

**Justification For Why Not Lower Score:**

N/A

---

### Decision · Program_Chairs · 2024-01-16

Reject